 Select

# Boundary criticality of the $O(N)$ model in $d = 3$ critically revisited

**Max A. Metlitski**

Department of Physics, Massachusetts Institute of Technology,
Cambridge, MA 02139, USA

mmetlits@mit.edu

## Abstract

It is known that the classical $O(N)$ model in dimension $d > 3$ at its bulk critical point admits three boundary universality classes: the ordinary, the extra-ordinary and the special. For the ordinary transition the bulk and the boundary order simultaneously; the extra-ordinary fixed point corresponds to the bulk transition occurring in the presence of an ordered boundary, while the special fixed point corresponds to a boundary phase transition between the ordinary and the extra-ordinary classes. While the ordinary fixed point survives in $d = 3$, it is less clear what happens to the extra-ordinary and special fixed points when $d = 3$ and $N \geq 2$. Here we show that formally treating $N$ as a continuous parameter, there exists a critical value $N_c > 2$ separating two distinct regimes. For $2 \leq N < N_c$ the extra-ordinary fixed point survives in $d = 3$, albeit in a modified form: the long-range boundary order is lost, instead, the order parameter correlation function decays as a power of $\log r$. For $N > N_c$ there is no fixed point with order parameter correlations decaying slower than power law. We discuss several scenarios for the evolution of the phase diagram past $N = N_c$. Our findings appear to be consistent with recent Monte Carlo studies of classical models with $N = 2$ and $N = 3$. We also compare our results to numerical studies of boundary criticality in 2+1D quantum spin models.



# 1   Introduction.

Boundary critical behavior is a subject with a long history [1–3] that has attracted renewed attention recently, driven in part by connections to the physics of symmetry protected topological phases. [4–13] In the present paper we will revisit this subject in the context of the classical $O(N)$ model. Let us recall what is known about this problem.

As a prototypical lattice model consider

$$\beta H = -\sum_{\langle ij \rangle} K_{ij} \vec{S}_i \cdot \vec{S}_j. \tag{1}$$

Here $\vec{S}_i$ is a classical $O(N)$ spin. $K_{ij} > 0$ is a nearest neighbour coupling that is taken to be $K_1$ if both $i$ and $j$ belong to the surface layer and $K$ otherwise. For bulk dimension $d > 3$ the conventionally accepted phase diagram has the schematic shape shown in Fig. 1.[1] Let us define the parameter $\kappa = K_1/K$. For $\kappa$ smaller than a critical value $\kappa_c$ the bulk and the boundary order at the same temperature $K = K_c$. This boundary universality class is known

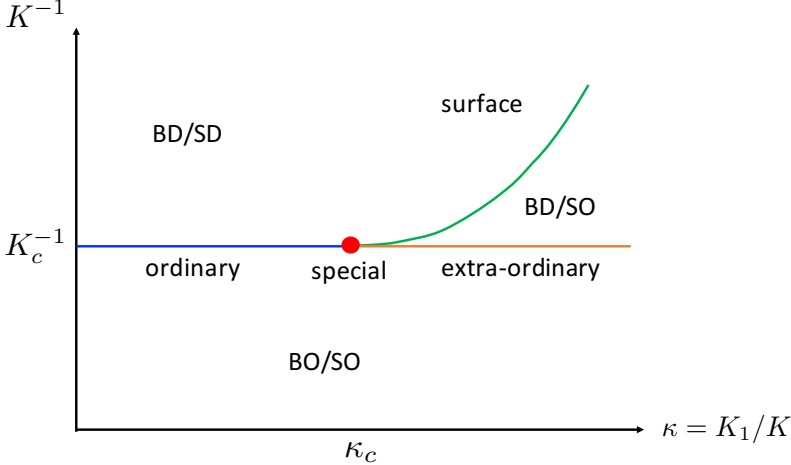

Figure 1: Conventionally accepted phase diagram of the classical $O(N)$ model with a boundary in dimension $d > 3$. BO stands for bulk ordered, SO - surface ordered, BD - bulk disordered, SD - surface disordered. For $d = 3$ and $N = 1$ the phase diagram is the same. For $d = 3$ and $N = 2$ the phase diagram has the same topology, but the BD/SO region only has quasi-long-range surface order.

---

[1]Here and below, we often formally treat variables $d$ and $N$ as continuous.

as "ordinary". On the other hand, for $\kappa > \kappa_c$ the enhancement of the surface coupling leads to the boundary ordering at a higher temperature than the bulk. Then for $\kappa > \kappa_c$ the onset of bulk order at $K = K_c$ in the presence of established boundary order is known as the "extra-ordinary" boundary universality class. Finally, the multicritical point at $\kappa = \kappa_c$ and $K = K_c$ is known as the "special" boundary universality class.

We note that the universality classes considered above correspond to no explicit symmetry breaking on the boundary. We can also consider the situation where one adds an explicit symmetry breaking field to the boundary $\delta H = -\sum_{i \in bound} \vec{h}_1 \cdot \vec{S}_i$. The boundary universality class at $K = K_c$ is then known as "normal." It is believed that for $d > 3$ the extra-ordinary universality class essentially coincides with the normal universality class. [14–17] Indeed, the presence of a finite boundary order parameter at the extra-ordinary transition effectively acts like a symmetry breaking field. This is most clear for the case of Ising spins ($N = 1$), but is also believed to be true for $N \geq 2$, where the Goldstone modes of the boundary effectively decouple from the bulk fluctuations at $K = K_c$.

Let's now turn our attention to dimension $d = 3$. For the case of Ising spins the boundary phase diagram remains the same as in Fig. 1. However, for $N \geq 2$ the situation is less clear - the present paper aims to shed light on precisely this question. For $N = 2$, the phase diagram has the same topology as in Fig. 1, however, now the region labelled as $BD/SO$ has only quasi-long-range boundary order rather than true long range order. [18–20] Then what happens if we start in this quasi-long-range ordered boundary phase and let $K$ approach $K_c$? (For simplicity, we will still refer to the ensuing transition as "extra-ordinary.") To our knowledge this question is not settled in the literature either analytically or numerically.[2] One possibility that has been discussed in the numerical study of Ref. [22] is that right at $K = K_c$ the boundary has true long range order, i.e. the boundary order parameter has a jump from 0 at $K < K_c$ to a finite value at $K > K_c$. This possibility cannot be immediately ruled out. Indeed, while for $K < K_c$ the boundary cannot develop true long range order by the Mermin-Wagner theorem, at $K = K_c$ the bulk effectively induces long-range interactions on the boundary that can lead to true long range order. In the present work we will use renormalization group (RG) to show that, in fact, this scenario is *not* realized in the $O(2)$ model in $d = 3$. Instead, we find that at $K = K_c$ for $\kappa > \kappa_c$ the order parameter correlation function on the boundary falls off as

$$\langle \vec{S}_x \cdot \vec{S}_0 \rangle \sim \frac{1}{(\log |\mathrm{x}|)^q} \, , \tag{2}$$

with $q$ - a universal exponent. Thus, the boundary comes close to ordering at $K = K_c$, but does not quite do so.[3] Further, as $K \to K_c^-$ the stiffness of the order parameter diverges logarithmically. Below, we will refer to this type of boundary critical behavior as "extra-ordinary-log".

Next, let's ask what happens in $d = 3$ for $N > 2$? Again, to our knowledge, this question is not settled in the literature. Now for $K < K_c$ the correlation length on the boundary is finite. Thus, the topology of the phase diagram does not mandate the existence of a separate "extra-ordinary" phase transition. Nevertheless, it is not ruled out that at $K = K_c$ there is a critical $\kappa_c$ separating two different boundary universality classes, even though these connect to the same paramagnetic phase for $K < K_c$, see Fig. 2, left.[4] In fact, if we treat $N$ as a continuous parameter, continuity would suggest that for $N$ just above 2 the extra-ordinary universality class survives. Our RG analysis supports this conclusion. On the other hand, for large (but finite) $N$ one would suspect that only the ordinary universality class remains, Fig. 2, right. Indeed, at $N = \infty$ one only finds an ordinary fixed point and no special fixed point. [14]

---

[2]In section 5, we will discuss the very recent numerical study, Ref. [21], which had appeared after the first arXiv version of the present paper.

[3]This is reminiscent of the behavior in the "Goldstone phase" of the 2d $O(N)$ model with $N < 2$. [23–25]

[4]In fact, examples of two regions of the phase boundary having different universality classes are also known for bulk phase transitions. [26,27]

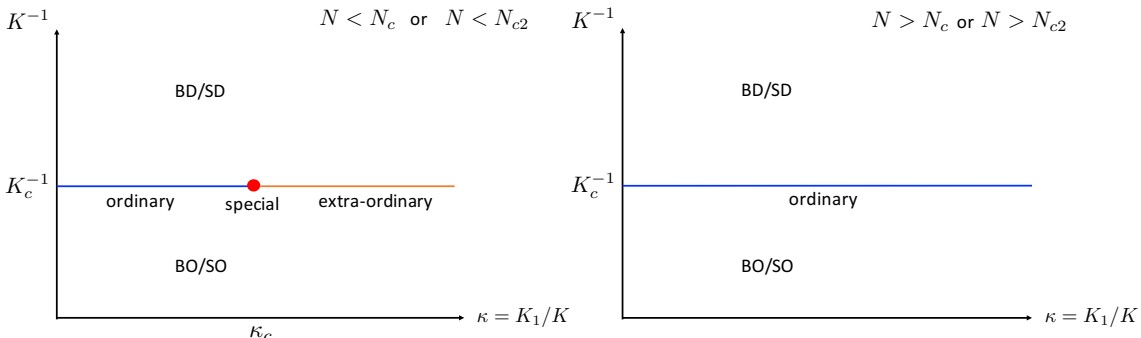

Figure 2: Proposed phase diagram of the classical $O(N)$ model for $d = 3$ and $N > 2$. Left: $N < N_c$ in scenario I or $N < N_{c2}$ in scenario II. Right: $N > N_c$ in scenario I or $N > N_{c2}$ in scenario II.

Further, using the large-$N$ results of Ref. [28] and setting the dimension $d = 3 + \epsilon$, $\epsilon > 0$, one finds that the boundary scaling dimension of the order parameter at the special fixed point is

$$\Delta_{\hat{\phi}}^{spec} \approx \epsilon \left( 1 + \frac{3}{N} \right) + O\left( \frac{1}{N^2} \right). \tag{3}$$

This suggests that for large but finite $N$ as $\epsilon \to 0$, $\Delta_{\hat{\phi}}^{spec}$ approaches zero, i.e. the special and the extra-ordinary fixed points approach each other and annihilate when $d = 3$. Again, our RG analysis confirms this.[5]

Returning to $d = 3$, taken together, the above findings about the behavior of the system for $N \to 2^+$ and for large but finite $N$ suggest that there exists a critical value of $N$, $N_c > 2$, separating two regimes.[6] For $2 \leq N < N_c$, the boundary behavior at $K = K_c$ is qualitatively the same as at $N = 2$, with an ordinary region, an extra-ordinary-log region where the surface order parameter correlation function falls of as (2), and a special fixed point separating them. For $N > N_c$ there is no extra-ordinary fixed point with logarithmically decaying correlations. In fact, there are two scenarios for the evolution of the system past $N_c$. In scenario I, Fig. 3 left, as $N \to N_c^-$ the the special and extra-ordinary fixed points "approach" each other, $\Delta_{\hat{\phi}}^{spec} \sim N_c - N$,

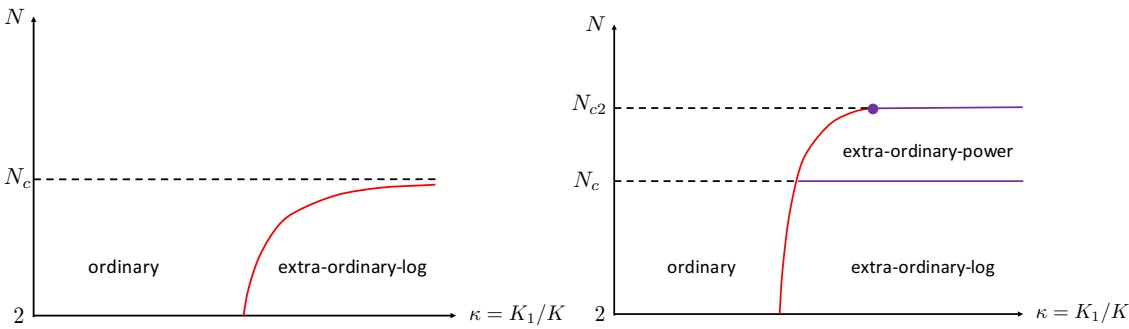

Figure 3: Proposed phase diagram of the classical $O(N)$ model for $d = 3$ and $N \geq 2$ at $K = K_c$. Left: scenario I. Right: scenario II. The dashed lines are a guide to eye and *do not* denote phase transitions. Solid lines are phase transitions. The red curve marks the special transition.

---

[5]The possibility of accessing the special fixed point in $d = 3 + \epsilon$ in a systematic expansion in $\epsilon$ was briefly speculated upon in Ref. [29].

[6]Note that $N_c$ is almost certainly not an integer.

and annihilate when $N = N_c$, so that for $N \geq N_c$ only the ordinary boundary universality class remains. In scenario II, Fig. 3 right, for $N$ just above $N_c$ both the ordinary and the extra-ordinary regions of the phase diagram and the special fixed point separating them remain, however, the correlation function of the order parameter in the extra-ordinary region now falls off as a power of $r$, $\Delta_{\hat\phi}^{ext} \sim N - N_c$. We will refer to this universality class as extra-ordinary-power. Since only the ordinary universality class exists at $N = \infty$, there should then be a second critical value $N_{c2} > N_c$, such that the special and extra-ordinary-power fixed points approach each other and annihilate as $N \to N_{c2}^-$. We note in passing that, as explained in section 4, scenario II also leads to a modification of the conventionally accepted phase diagram in Fig. 1 for $d$ just above 3.

Unfortunately, at the present time we do not precisely know the exact value of $N_c$. Our RG analysis allows $N_c$ to be determined from the knowledge of certain universal amplitudes at the "normal" fixed point (which exists for all $N$ in $d = 3$). However, these amplitudes are not known exactly. From the large-$N$ expansion, we estimate $N_c \approx 4$, however, it is not clear how accurate this estimate is. Further, we currently don't know which of the two scenarios above for the evolution of the system for $N > N_c$ is realized. This is ultimately determined by a sign of a higher order term in a certain $\beta$-function in our RG analysis, which at present we are not able to compute. Numerical simulations can yield answers to these questions. In sections 5, 7, we will discuss the existing Monte Carlo data on the classical $O(N)$ model and on quantum spin models in 2+1D with $SO(3)$ symmetry. In particular, a recent Monte Carlo study of the classical $O(3)$ model [30] that has appeared after the first arXiv version of the present paper, finds a special transition at a critical value $\kappa_c$ and behavior at large $\kappa$ compatible with the extra-ordinary-log boundary universality class. This implies that $N_c > 3$.

This paper is organized as follows. In section 2 we develop a field theoretic formalism to study the boundary critical behavior. Our treatment is essentially an expansion around the ordered boundary state combined with renormalization group. It is similar in spirit to the RG treatment of the non-linear $\sigma$-model strictly in $d = 2$; our progress on the boundary problem is enabled by the fact that some critical exponents for the normal universality class are known exactly. [14–17] In section 3 we derive the renormalization group equations, and section 4 is devoted to an analysis of their consequences for the phase diagram. Section 5 compares our results to existing Monte Carlo data on the classical $O(N)$ model, particularly to the recent large scale studies that observe behavior consistent with the extra-ordinary-log phase in the large $\kappa$ region of the $N = 2$ [21] and $N = 3$ [30] models. Section 6 is devoted specifically to the case $N = 2$. Here we discuss both classical models in (bulk) $d = 3$ and quantum models in $D = 2 + 1$ (e.g. the Bose-Hubbard model). We comment on the role of vortices at the special transition. In particular, for the quantum model in the case when the boundary boson density $\rho$ is incommensurate, such that phase slips on the boundary are prohibited by translational symmetry, we are able to describe not only the extra-ordinary-log phase, but also the special fixed point separating it from the ordinary boundary universality class, see Fig. 6. Section 7 is devoted to quantum models with $N = 3$. Here we review recent numerical results on 2+1D quantum spin models and discuss their possible theoretical interpretation. Some concluding remarks are presented in section 8.

## 2 Set-up.

Consider the classical lattice model (1) in $d$-dimensions and let the boundary be at $x_d = 0$. We will be mostly interested in the case $d = 3$, but will keep $2 < d < 4$ general for now. We wish to study this model at its bulk critical point $K = K_c$ and in the $\kappa \gg 1$ region, when the surface has a strong tendency to local order. First, imagine turning off the couplings connecting the

outermost surface layer to the next layer. The system without the outermost surface layer is then expected to realize the *ordinary* boundary universality class. We call the corresponding continuum fixed-point bulk+boundary action of the $d$-dimensional $O(N)$ model, $S_{ordinary}$. We denote the bulk order parameter of the $O(N)$ model by $\vec{\phi}(\mathrm{x}, x_d)$. The initially decoupled outermost surface layer can be described by the $d-1$ dimensional continuum $O(N)$ non-linear $\sigma$-model for the field $\vec{n}$,

$$S_n = \int d^{d-1}\mathrm{x} \left( \frac{1}{2g}(\partial_\mu \vec{n})^2 - \vec{h} \cdot \vec{n} \right), \quad \vec{n}^2 = 1. \tag{4}$$

Here, $\vec{h}$ is a small symmetry breaking field that will be used as an infra-red regulator. When $\kappa \gg 1$, we expect $g$ to be small.

Now, let's restore the coupling of the outermost surface layer to the next layer: in the continuum description, we expect a coupling

$$S_{n\phi} = -\tilde{s} \int d^{d-1}\mathrm{x}\, \vec{n}(\mathrm{x}) \cdot \vec{\phi}(\mathrm{x}, x_d = 0) \tag{5}$$

to be generated. Here $\vec{\phi}(\mathrm{x}, x_d = 0)$ should be understood as the lowest dimension $O(N)$ vector boundary operator of the $O(N)$ model at its ordinary boundary fixed point. Thus, we study the action

$$S_{UV} = S_{ordinary} + S_n + S_{n\phi}. \tag{6}$$

We want to understand what are the effects of the coupling $\tilde{s}$. To do so, we will work around the fixed point $g = 0$. When $g$ is strictly zero, the fluctuations of $\vec{n}$ are frozen. Let's choose $\vec{n}$ to point along the $N$-th direction. The coupling $S_{n\phi}$ then acts as a boundary symmetry breaking field for the bulk $O(N)$ model. Such a field is relevant at the ordinary boundary fixed point and makes the boundary flow to the so-called "normal" fixed point.

A lot is known about the normal fixed point. First of all, this fixed point exists for all $N$ and $d > 2$. Second, the bulk order parameter acquires a finite expectation value near the boundary. Thus, if the symmetry breaking field points along the $N$-th direction, letting $\phi_N = \sigma$, we have the operator product expansion (OPE):

$$\sigma(\mathrm{x}, x_d) \sim \frac{a_\sigma}{(2x_d)^{\Delta_\phi}} + b_{\mathrm{D}}(2x_d)^{d-\Delta_\phi}\hat{\mathrm{D}}(\mathrm{x}) + \dots, \quad x_d \to 0. \tag{7}$$

Here, $a_\sigma$ and $b_{\mathrm{D}}$ are universal constants.[7] Note, we normalize the bulk operators so that in the absence of the boundary $\langle O^a(x)O^b(y)\rangle = \frac{\delta^{ab}}{|x-y|^{2\Delta_O}}$. Likewise for the boundary operators, $\langle \hat{O}^a(\mathrm{x})\hat{O}^b(\mathrm{y})\rangle = \frac{\delta^{ab}}{|\mathrm{x}-\mathrm{y}|^{2\Delta_{\hat{O}}}}$. We generally denote boundary operators with a hat. Besides the identity, the lowest dimension field $\hat{\mathrm{D}}(\mathrm{x})$ contributing to the OPE on the RHS of (7) (i.e. the lowest dimension $O(N-1)$ scalar) is believed to be the "displacement" operator, which has scaling dimension of exactly $d$. [15, 31] Thus, since $\Delta_{\hat{\mathrm{D}}} > d-1$, the normal fixed point has no relevant boundary perturbations that don't break the remnant $O(N-1)$ symmetry. The boundary scaling dimension of the lowest $O(N-1)$ vector on the boundary $\hat{\mathrm{t}}_i$, $i = 1 \dots N-1$ is also known exactly: $\Delta_{\hat{\mathrm{t}}} = d-1$. [14, 31, 32][8] We write:

$$\phi_i(\mathrm{x}, x_d) \sim b_{\mathrm{t}}(2x_d)^{d-1-\Delta_\phi}\hat{\mathrm{t}}_i(\mathrm{x}) + \dots, \quad x_d \to 0, \tag{8}$$

where $b_{\mathrm{t}}$ is a universal constant.

---

[7]We are using the normalization convention of Ref. [31], which differs from the convention used in the first arXiv version of this paper.

[8]The label "t" stands for "tilt", a term coined in Ref. [33]

While the action (6) provides a conceptually clear $O(N)$ symmetric regularization of the model we wish to consider, it is inconvenient to work with. Indeed, even at $g = 0$ we don't know the details of the flow from the ordinary to the normal boundary fixed point of the $O(N)$ model. Thus, we'd like to start with the end-point of this flow. We consider

$$S_{IR} = S_{normal} + S_n - s \int d^{d-1}\mathrm{x}\, \pi_i(\mathrm{x})\hat{\mathrm{t}}_i(\mathrm{x}) + \delta S\,, \tag{9}$$

where $\vec{n} = (\vec{\pi}, \sqrt{1-\vec{\pi}^2})$ and $S_{normal}$ is the conformal fixed point of the $O(N)$ model with a normal boundary (and the symmetry-breaking field pointing along the $N$th direction). $\hat{\mathrm{t}}_i$ is the boundary $O(N-1)$ vector at the normal fixed point, Eq. (8). Note that while the first three terms in $S_{IR}$ enjoy an $O(N-1)$ symmetry, they don't have an explicit $O(N)$ symmetry that our UV action (6) possesses. Indeed, the coupling of the $N$-th component of bulk and boundary $\phi$ field, $\delta L \sim n_N \cdot \hat{\mathrm{D}}$ is irrelevant in the RG sense at the $g = 0$ fixed point and so will not be included. Thus, the action (9) must somehow have an emergent $O(N)$ symmetry. Another comment is that the coupling $s$ in (9) is actually not the same as the coupling $\tilde{s}$ in the UV action (5). Indeed, $s$ is the effective coupling emerging after the RG flow from the ordinary to the normal fixed point. In fact, we will see momentarily that in order to have $O(N)$ symmetry, $s$ will be fixed at a particular value. Finally, the term $\delta S$ consists of counter-terms that we will adjust order by order in $\pi$ (equivalently, $g$) to restore the $O(N)$ invariance.

## 2.1 Fixing the value of $s$.

We now use the $O(N)$ symmetry to fix the value of $s$.[9] We continue to work at $g = 0$, where $\vec{n}$ is a classical frozen constant field. When this field points along the $N$-th direction, $\vec{n} = (\vec{0}, 1)$ we know that

$$\langle \sigma(x_d) \rangle = \frac{a_\sigma}{(2x_d)^{\Delta_\phi}}\,, \quad \langle \phi_i \rangle = 0\,. \tag{10}$$

If we rotate $\vec{n}$ by an infinitesimal angle $\alpha$ towards the direction $\hat{1}$, $n_1 = \sin\alpha, n_N = \cos\alpha$, we should get

$$\langle \phi_1(x_d) \rangle = \frac{a_\sigma \sin\alpha}{(2x_d)^{\Delta_\phi}}\,. \tag{11}$$

But, from (9), to first order in $\alpha$,

$$\langle \phi_1(x_d) \rangle = s\alpha \int d^{d-1}\mathrm{x}\, \langle \phi_1(0, x_d)\hat{\mathrm{t}}_1(\mathrm{x}) \rangle_{norm}, \tag{12}$$

where the subscript $norm$ denotes the expectation value taken with respect to the action $S_{normal}$. The correlation function on the RHS is fixed by conformal symmetry. [34] Indeed, we have

$$\langle \phi^i(\mathrm{x}, x_d)\phi^j(\mathrm{x}', x_d') \rangle_{norm} = \frac{\delta^{ij}}{(4x_d x_d')^{\Delta_\phi}} g(\xi), \quad \xi = \frac{|\mathrm{x}-\mathrm{x}'|^2 + (x_d - x_d')^2}{4x_d x_d'}\,, \tag{13}$$

with $g(\xi)$ - a universal function. Our choice of normalization of $\phi^i$ in the absence of the boundary implies $g(\xi) \to \xi^{-\Delta_\phi}$, $\xi \to 0$. Further, using the OPE (8) on both operators in the correlator (13) requires

$$g(\xi) \to \frac{b_t^2}{\xi^{d-1}}\,, \quad \xi \to \infty\,. \tag{14}$$

---

[9]In fact, this is the same argument that is used to fix $\Delta_{\hat{\mathrm{t}}} = d - 1$. [14]

Now, using the OPE (8) on just one of the operators in (13) we obtain,

$$\langle \phi_i(\mathbf{x}, x_d) \hat{t}_j(\mathbf{x}') \rangle_{norm} = b_t \delta_{ij} \frac{(2x_d)^{d-1-\Delta_\phi}}{(|\mathbf{x} - \mathbf{x}'|^2 + x_d^2)^{d-1}} . \tag{15}$$

Substituting this into (12) and taking the integral over x,

$$s = \frac{\Gamma(d-1)}{(4\pi)^{\frac{d-1}{2}} \Gamma(\frac{d-1}{2})} \frac{a_\sigma}{b_t} \overset{d=3}{=} \frac{1}{4\pi} \frac{a_\sigma}{b_t} . \tag{16}$$

Thus, the coupling $s$ is fixed by the $O(N)$ symmetry in terms of the universal constants $a_\sigma$ and $b_t$. Crucially, $s$ is dimensionless. We also note that $s$ is not small. Thus, we will not be performing perturbation theory in $s$, but rather in $g$.

We note that so far we've only restored the $O(N)$ symmetry to leading order in fluctuations of $\vec{n}$ (in particular, the above analysis was carried out to linear order in $\alpha$ only). We may need to add extra terms to the Lagrangian to restore the symmetry at higher orders. One example of such a term is $\delta L \sim \vec{\pi}^2 \pi_i \hat{t}_i$. However, these higher order terms will not affect our analysis below.

The value of $s$ plays an important role in what follows, so we pause to discuss various results for $a_\sigma$ and $b_t$. Explicit expressions can be obtained for $a_\sigma$ and $b_t$ in the limit $N \to \infty$. We have computed the first corrections to these quantities in $1/N$ (see appendix A) for $d = 3$. (In fact, all the steps in the computation were already explained in Ref. [35], however, no explicit final result was given, so we repeat the calculation here.)

$$
\begin{aligned}
a_\sigma^2 &= 2(N+1)\left(1 - \frac{\eta}{2}\right) + O\left(\frac{1}{N}\right) \approx 2(N + 0.865) + O\left(\frac{1}{N}\right), \\
b_t^2 &= \frac{1}{4}\left(1 + \frac{1}{N}\right)\left(1 - \frac{\eta}{2}\right) + O\left(\frac{1}{N^2}\right) \approx \frac{1}{4}\left(1 + \frac{0.865}{N}\right) + O\left(\frac{1}{N^2}\right), \\
s^2 &= \frac{N}{2\pi^2} + O\left(\frac{1}{N}\right).
\end{aligned}
\tag{17}
$$

Here $\eta \approx \frac{8}{3\pi^2 N}$ is the bulk anomalous dimesnion of $\phi$: $\Delta_\phi = (d - 2 + \eta)/2$.

One can also obtain expressions for $a_\sigma^2$ and $b_t^2$ in the $4 - \epsilon$ expansion from the results of Ref. [36]:

$$a_\sigma^2 = \frac{4(N+8)}{\epsilon}\left(1 - \frac{N^2 + 31N + 154}{(N+8)^2}\epsilon\right), \tag{18}$$

$$b_t^2 = \frac{1}{3}\left(1 - \frac{N+9}{6(N+8)}\epsilon\right). \tag{19}$$

Unfortunately, the utility of Eqs. (18),(19), in $d = 3$ is not clear. In fact, substituting $\epsilon = 1$ into (18) gives a negative $a_\sigma^2$ for all $N$. On the other hand, substituting $\epsilon = 1$ into (19) gives $b_t^2$ within 15% of the large-$N$ estimate (17) for $N \geq 3$.

## 3 RG.

We now perform RG on the model (9). Since the coupling $s$ has been fixed by symmetry, only the coupling $g$ is allowed to run. As in the standard $O(N)$ model near 2d we let

$$g = \mu^{-\epsilon} Z_g g_r, \qquad \vec{n} = Z_n^{1/2} \vec{n}_r, \tag{20}$$

where the bulk dimension $d = 3 + \epsilon$, $\mu$ is the RG scale, $\Lambda$ is the UV cut-off, $g_r$ is the renormalized dimensionless coupling and $Z_g, Z_n$ are functions of $g_r$ and $\Lambda/\mu$. (We will be mostly interested

in the behavior in $d = 3$, however, it will occasionally be useful to consider $d = 3 + \epsilon$ to compare our results to those known in the literature.) The $\beta$-function and the the anomalous dimension are defined as

$$\beta(g_r) = \mu \frac{\partial g_r}{\partial \mu}\bigg|_{g,\Lambda}, \quad \eta_n(g_r) = \mu \frac{\partial}{\partial \mu} \log Z_n\bigg|_{g,\Lambda}. \tag{21}$$

The renormalized $m$-point function of the $\vec{n}$ field, $D_r^m = Z_n^{-m/2}\langle n(x_1)n(x_2)\dots n(x_m)\rangle$, then satisfies for $\vec{h} = 0$,

$$\left(\mu \frac{\partial}{\partial \mu} + \beta(g_r)\frac{\partial}{\partial g_r} + \frac{m}{2}\eta_n(g_r)\right)D_r^m(g_r, \mu) = 0. \tag{22}$$

We can extract $Z_g$ and $Z_n$ by requiring that the two-point function $\langle \pi_r^i(x)\pi_r^j(y)\rangle$ and the one-point function $\langle n_r^N(x)\rangle$ be independent of $\Lambda$. In the absence of the coupling to the bulk fields this gives to leading non-trivial order in $g_r$ (and to zeroth order in $\epsilon$):

$$Z_n = 1 - \frac{N-1}{2\pi}g_r \log\frac{\Lambda}{\mu}, \qquad Z_g = 1 - \frac{N-2}{2\pi}g_r \log\frac{\Lambda}{\mu}, \tag{23}$$

so that [37]:

$$\beta(g_r) = \epsilon g_r - \frac{N-2}{2\pi}g_r^2, \qquad \eta_n(g_r) = \frac{N-1}{2\pi}g_r. \tag{24}$$

Now, let's include the effect of the coupling $s$ to the bulk fields. To leading order in $g$, $\langle n_N\rangle \approx \langle 1 - \frac{1}{2}\pi^2\rangle$ is unmodified. Thus, $Z_n$ remains unmodified to this order. However, the two point function $\langle \pi_i(x)\pi_j(x)\rangle$ receives an extra contribution:

$$\delta_s\langle \pi_i(x)\pi_j(y)\rangle = s^2 \int d^{d-1}z\, d^{d-1}w\, D_0(x,z)\langle \hat{t}_i(z)\hat{t}_j(w)\rangle_{norm}D_0(w,y). \tag{25}$$

Here,

$$D_0(x,y) = \int \frac{d^{d-1}p}{(2\pi)^{d-1}}\frac{g}{p^2 + gh}e^{i\vec{p}\cdot(\vec{x}-\vec{y})} \tag{26}$$

is the bare $\pi$ propagator. We will denote the full $\pi$ propagator by $D$. Going to momentum space and using the normalization of $\hat{t}$ operator:

$$\delta_s D(p) = s^2 D_0^2(p) \int d^{d-1}z\, \frac{1}{|z|^{2(d-1)}}e^{-i\vec{p}\cdot\vec{z}}. \tag{27}$$

Alternatively, letting the self-energy $\Sigma_\pi(p)$ be defined as $D(p)^{-1} = D_0(p)^{-1} + \Sigma_\pi(p)$,

$$\delta_s\Sigma_\pi(p) = -s^2 \int d^{d-1}z\, \frac{1}{|z|^{2(d-1)}}e^{-i\vec{p}\cdot\vec{z}}. \tag{28}$$

We notice that $\delta_s\Sigma_\pi(p = 0, h = 0) \neq 0$. This would lead to the breaking of $O(N)$ rotational symmetry. Thus, to restore the $O(N)$ symmetry, we add a counterterm $\delta S = C \int d^{d-1}x\, \vec{\pi}^2(x)$, with $C = -\delta_s\Sigma_\pi(p = 0)$. Following this,

$$\delta_s\Sigma_\pi(p) \to -s^2 \int d^{d-1}z\, \frac{1}{|z|^{2(d-1)}}\left(e^{-i\vec{p}\cdot\vec{z}} - 1\right). \tag{29}$$

Setting $d = 3$ and performing the integral we obtain to logarithmic accuracy,

$$\delta_s\Sigma_\pi(p) = \frac{\pi s^2}{2}p^2 \log\frac{\Lambda}{p}. \tag{30}$$

Therefore, to eliminate this divergence we choose

$$Z_g = Z_g^{O(n)}\left(1 + \frac{\pi s^2}{2} g_r \log \frac{\Lambda}{\mu}\right) = 1 - \left(\frac{N-2}{2\pi} - \frac{\pi s^2}{2}\right) g_r \log \frac{\Lambda}{\mu}, \tag{31}$$

with $Z_g^{O(n)}$ given by Eq. (23). Therefore,

$$\beta(g_r) = \epsilon g_r - \left(\frac{N-2}{2\pi} - \frac{\pi s^2}{2}\right) g_r^2, \qquad \eta_n(g_r) = \frac{N-1}{2\pi} g_r. \tag{32}$$

## 4 Phase diagram.

What are the consequences of the RG analysis in section 3? Letting $\ell$ be the RG scale, $\mu x \sim e^\ell$, we have $\frac{dg_r}{d\ell} = -\beta(g_r)$. Writing $\beta(g_r) = \epsilon g_r + \alpha g_r^2$ with

$$\alpha = \frac{\pi s^2}{2} - \frac{N-2}{2\pi}, \tag{33}$$

we observe that the physics depends on the sign of $\alpha$. When $\alpha > 0$ the $g = 0$ fixed point is stable in $d = 3$, while when $\alpha < 0$ it is unstable. Crucially, we known that when $N = 2$,

$$\alpha(N = 2) = \frac{\pi s^2}{2} > 0, \tag{34}$$

while for large (but finite $N$), from Eq. (17)

$$\alpha(N) \approx -\frac{(N-4)}{4\pi} + O\left(\frac{1}{N}\right) < 0, \quad N \to \infty. \tag{35}$$

Thus, there must be a critical $N_c$ at which $\alpha$ switches sign.[10] Naive extrapolation of the large-$N$ result (35) gives $N_c \approx 4$.

Let's begin our analysis in $d = 3$ with $2 \leq N < N_c$. Here $g_* = 0$ is a stable fixed point as $g$ runs logarithmically to zero:

$$g_r(\ell) = \frac{g_r}{1 + \alpha g_r \ell}, \quad d = 3. \tag{36}$$

Further, integrating the Callan-Symanczyk equation,

$$\langle n_N \rangle_r \sim \left(1 + \alpha g_r \log \frac{\mu}{\sqrt{h}}\right)^{-q/2} \to 0, \quad h \to 0, \tag{37}$$

where

$$q = \frac{N-1}{2\pi\alpha}. \tag{38}$$

The order parameter expectation value vanishes as a power of logarithm as $h \to 0$, thus, unlike for $d > 3$, there is no true long range order at the $g_* = 0$ fixed point in $d = 3$. Further, the two point function of $\vec{n}$ for $h = 0$,

$$D_r(p) \approx \frac{g_r}{p^2 \left(1 + \alpha g_r \log \frac{\mu}{p}\right)^{1+q}}. \tag{39}$$

---

[10]Here and below we assume the minimal scenario where $\alpha(N)$ has only a single zero for $N \geq 2$.

Integrating this, we obtain the propagator in space to leading logarithmic accuracy:

$$D(\mathrm{x}) \propto \frac{1}{(\log \mu \mathrm{x})^q}, \quad \mathrm{x} \to \infty. \tag{40}$$

Thus, for $2 \leq N < N_c$, the extra-ordinary fixed point survives in $d = 3$, but in a modified form with no true long range order and order parameter correlations that decay only as a power of logarithm. We refer to such behavior as the extra-ordinary-log universality class. In particular, this occurs for the case $N = 2$ where we are confident that $\alpha > 0$. Here, the extra-ordinary-log fixed point that we've just obtained controls the behavior as we approach $K = K_c$ out of the surface phase with quasi-long-range order, labelled as BD/SO in Fig. 1. Calling the bulk correlation length $\xi_{bulk} \sim (K_c - K)^{-\nu_{bulk}}$, we will have the flow (36) cut-off at $\ell \sim \log(\mu \xi_{bulk})$. Thus,

$$\mathcal{K} = \frac{\pi}{g(\ell)} \sim \frac{\pi}{g_r} + \pi \alpha \log(\mu \xi_{bulk}), \tag{41}$$

where $\mathcal{K}$ is the Luttinger parameter of the surface superfluid. We see that the Luttinger parameter diverges logarithmically as $K \to K_c^-$.

While the existence of an extra-ordinary transition in $d = 3$ for $N = 2$ was mandated by the topology of the phase diagram, it is more curious that this transition survives for $2 < N < N_c$, where for $K < K_c$ all surface correlators decay exponentially and no extra-ordinary transition is required. Of course, current analytic understanding does not tell us whether $N_c > 3$, i.e. whether the above range includes any integer values of $N$ (see, however, the discussion of recent Monte Carlo results in section 5). Still, formally in this range the approach to $K_c$ in the region $\kappa > \kappa_c$ in Fig. 2, left, is controlled by the extra-ordinary-log fixed point. In this case, the flow in Eq. (36) is again cut-off at $\ell \sim \log(\mu \xi_{bulk})$, after which the flow controlled by the strictly 2d $O(N)$ $\beta$-function (24) resumes. This gives a surface correlation length

$$\frac{\xi_{surf}}{\xi_{bulk}} \sim (\xi_{bulk})^{\frac{2\pi \alpha}{N-2}}, \tag{42}$$

i.e. the surface correlation length is parametrically larger than the bulk correlation length as $K \to K_c^-$.

We note that in the range $2 \leq N < N_c$ since a stable extra-ordinary fixed point $g_* = 0$ exists, there must also be a special fixed point separating the extra-ordinary and the ordinary fixed points. However, for a general $N$ in this range the special fixed point does not occur at a parametrically weak coupling $g$. What happens for $N$ close to $N_c$? The physics here will be controlled by the sign of the cubic term in the $\beta$-function:

$$\beta(g) \approx \alpha g^2 + b g^3. \tag{43}$$

For $N \to N_c$, writing $\alpha \approx a(N_c - N)$ with $a > 0$, we need to know the sign of $b(N_c)$. At present we do not know this sign,[11] so we consider both scenarios.

- Scenario I: $b(N_c) < 0$. Then for $N$ just below $N_c$, we have two perturbatively accessible fixed points: the stable fixed point $g_* = 0$ corresponding to the extra-ordinary-log class and the unstable fixed point $g_*^{spec} \approx \frac{a(N_c - N)}{|b|}$. We guess that $g_*^{spec}$ corresponds to the special transition and the flow $g \to \infty$ for $g > g_*^{spec}$ to the flow to the ordinary fixed point. Then as $N \to N_c^-$, the special fixed point approaches the extra-ordinary fixed point and annihilates with it. At the special fixed point the scaling dimension of the surface order parameter

$$\Delta_{\vec{n}}^{spec} = \frac{\eta_n(g_*^{spec})}{2} \approx \frac{(N-1)g_*^{spec}}{4\pi} \tag{44}$$

---

[11] We note that for the pure $d = 2$ $O(N)$ model $b(N) < 0$ [38].

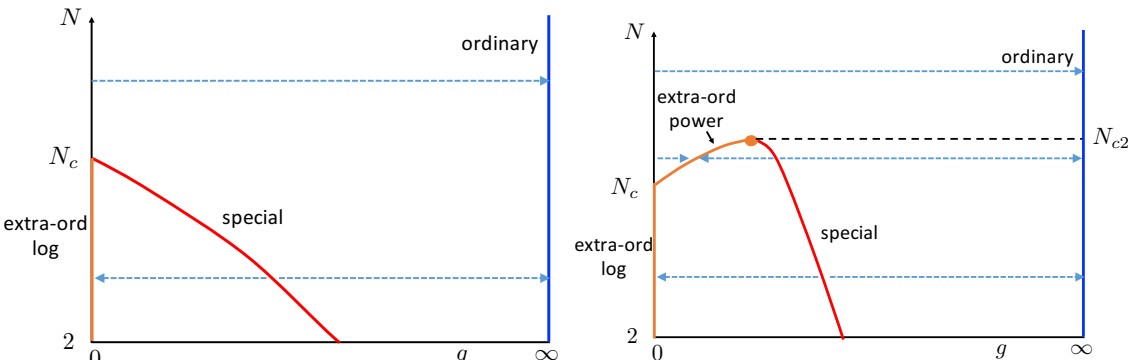

Figure 4: Conjectured RG flows as a function of $N$ in $d = 3$. Left - scenario I. Right - scenario II. Blue dashed arrows indicate the direction of RG flow. Black dashed lines are guide to eye.

will then be small for $N$ just below $N_c$. For $N > N_c$ any finite $g > 0$ flows to $g = \infty$ signifying that only the ordinary fixed point is left and we have the phase diagram in Fig. 2, right. The overall evolution of fixed points as a function of $N$ in this scenario is sketched in Fig. 4, left.

- Scenario II: $b(N_c) > 0$. In this scenario, the special fixed point *does not* approach $g = 0$ as $N \to N_c$. Rather, the $g = 0$ fixed point becomes unstable for $N > N_c$, but there is a stable, perturbatively accessible fixed point at $g_*^{ext-p} \approx \frac{a(N-N_c)}{b}$. Thus, for $N$ just above $N_c$ the phase diagram still has the topology in Fig. 2 left, but the extra-ordinary transition is now controlled by the new fixed point $g_*^{ext-p}$ and the surface order parameter correlations at it have a power-law behavior. We refer to this universality class as extra-ordinary-power. As $N$ increases further, we expect that eventually the $g_*^{ext-p}$ fixed point approaches the special fixed point and annihilates with it at $N = N_{c2} > N_c$. Then for $N > N_{c2}$ only the ordinary universality class remains, as expected from large-$N$ expansion. The overall evolution of fixed points as a function of $N$ in this scenario is sketched in Fig. 4, right. While the phase diagram in this scenario is more complicated than in Scenario I, at present we cannot rule Scenario II out.

Before we conclude this section, we present the conjectured RG flow of the model in

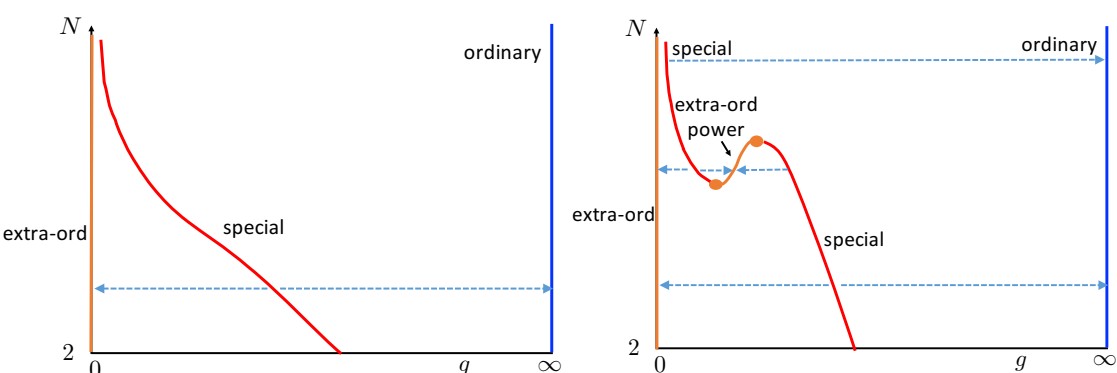

Figure 5: Conjectured RG flows as a function of $N$ in $d = 3 + \epsilon$. Left - scenario I. Right - scenario II. Blue dashed arrows indicate the direction of RG flow. In scenario II, as $\epsilon$ increases the figure on the right will eventually evolve into the figure on the left.

$d = 3 + \epsilon$ with $\epsilon \ll 1$. Fig. 5, left, shows the RG flow in scenario I and Fig. 5, right - in scenario II. First of all, the $g = 0$ fixed point is stable in $d > 3$ and corresponds to the conventional extra-ordinary transition, where the boundary orders before the bulk. Indeed, $\langle \vec{n} \rangle$ has a finite expectation value at this fixed point. Further, the fact that $g(\ell) \sim g e^{-\epsilon \ell}$ flows to 0 at this fixed point implies that the fluctuations of $\vec{n}$ essentially decouple from the bulk with normal boundary conditions: this is the known equivalence of the extra-ordinary and normal transitions in $d > 3$. We further note that fixing $N > N_c$ and letting $\epsilon \to 0$, we have an additional perturbatively accessible unstable fixed point at $g_*^{spec,1} = \frac{\epsilon}{|\alpha|}$. For $N \gtrsim N_c$ in scenario I and $N \gtrsim N_{c2}$ in scenario II $g_*^{spec,1}$ separates the extra-ordinary and the ordinary phases and corresponds to the special transition in the conventional phase diagram of Fig. 1. Note that in scenario II, we also have a region $N_c \lesssim N \lesssim N_{c2}$ where there are three stable phases: the extra-ordinary, the extra-ordinary-power and the ordinary and two successive transitions separating them (with $g_*^{spec,1}$ being the one at the smaller value of $g$). Such a region was not previously foreseen. Since in $d = 4 - \epsilon$ we only have the extra-ordinary and the ordinary stable phases, we conclude that if Scenario II is realized, Fig. 5, right, must evolve into Fig. 5, left, as $d$ increases from 3 to 4.

A highly non-trivial check of our RG analysis is obtained by comparing the behavior at the special transition $g_*^{spec,1}$ in $d = 3 + \epsilon$ for large $N$ to what is known from direct large-$N$ treatment of the special transition. From (35), we have

$$g_*^{spec,1} \approx \frac{4\pi\epsilon}{N}\left(1 + \frac{4}{N} + O(N^{-2})\right), \quad \eta_n^{spec,1} \approx 2\epsilon\left(1 + \frac{3}{N} + O(N^{-2})\right), \quad N \to \infty. \quad (45)$$

Then, at $g_*^{spec,1}$ from the Callan-Symanczyk equation (22), $\langle n^i(\mathrm{x}) n^j(0) \rangle \sim \frac{\delta^{ij}}{\mathrm{x}^{\eta_n}}$, i.e

$$\Delta_{\vec{n}}^{spec,1} = \frac{\eta_n}{2} = \epsilon\left(1 + \frac{3}{N} + O(N^{-2})\right) + O(\epsilon^2). \quad (46)$$

On the other hand, from direct large-$N$ expansion in arbitrary $3 < d < 4$, the boundary scaling dimension of $\phi$ at the special transition is:[12]

$$\Delta_{\hat{\phi}}^{spec} = d - 3 + \frac{1}{N}\frac{2(4-d)}{\Gamma(d-3)}\left[\frac{(6-d)\Gamma(2d-6)}{d\Gamma(d-3)} + \frac{1}{\Gamma(5-d)}\right] + O\left(\frac{1}{N^2}\right), \quad (47)$$

which exactly matches Eq. (46) to first order in $\epsilon$ and to $O(1/N)$.

Another observation is that at $g_*^{spec,1}$ the boundary correlation length exponent,

$$\nu^{spec,1} = \frac{1}{|\beta'(g_*)|} = \frac{1}{\epsilon}. \quad (48)$$

This means that the dimension of the relevant $O(N)$ scalar boundary operator $\hat{O}_{rel}^{spec}$ that drives one away from the special critical point is $\Delta_{\hat{O}_{rel}^{spec}} = d - 1 - 1/\nu^{spec} = 2 + O(\epsilon^2)$. For $N = \infty$, this agrees with the scaling dimension of the lightest boundary $O(N)$ scalar - see Eq. (5.8) in Ref. [28]. Note that Eq. (48) is actually correct to leading order in $\epsilon$ for any $N$, as long as $N > N_c$.

## 4.1 Velocity running.

So far we've been thinking of classical models and assuming that there is sufficient rotational symmetry to guarantee isotropy of $(\nabla \vec{n})^2$. However, in quantum models there is no reason for

---

[12] We are using $\Delta_{\hat{\phi}} = \frac{d-2+\eta_\parallel}{2}$, with $\eta_\parallel$ given by Eq. 5.15a of Ref. [28].

the velocity of the bulk and boundary modes to be the same. In particular, we should modify our action to

$$S = S_{normal} + \frac{1}{2g} \int dx\, d\tau \left( \frac{1}{v_s}(\partial_\tau \vec{n})^2 + v_s(\partial_x \vec{n})^2 \right) - s v_b \int dx\, d\tau\, \pi_i \hat{t}_i \,. \tag{49}$$

Here we are taking bulk dimension to be $d = 2 + 1$. $v_s$ is the surface velocity and $v_b$ - the bulk velocity. $g$ is dimensionless and $s$ is again given by Eq. (16). We normalize $\hat{t}_i$ to have the correlation function

$$\langle \hat{t}_i(x,\tau) \hat{t}_j(0,0) \rangle = \frac{\delta_{ij}}{(x^2 + v_b^2 \tau^2)^2} \,. \tag{50}$$

Note that the bulk velocity $v_b$ is not renormalized by the surface. Then repeating the calculations in section 3 we obtain

$$\begin{aligned}
\frac{d(v_s/v_b)}{d\ell} &= -\frac{\pi s^2 g}{4} \left( \left( \frac{v_s}{v_b} \right)^2 - 1 \right), \\
\frac{dg}{d\ell} &= \left( \frac{N-2}{2\pi} - \frac{\pi s^2}{4} \left( \frac{v_s}{v_b} + \frac{v_b}{v_s} \right) \right) g^2 \,.
\end{aligned} \tag{51}$$

As expected, the running of $v_s$ vanishes when $v_s = v_b$. Further, the flow is towards $v_s = v_b$ whenever we have a perturbatively accessible fixed point for $g$. In particular, for the extraordinary-log fixed point, assuming $v_s/v_b$ is initially close to 1, we may integrate the RG equation for $g(\ell)$, obtaining Eq. (36). Then substituting this into the RG equation for $v_s/v_b$ we obtain

$$\frac{v_s(\ell)/v_b - 1}{v_s/v_b - 1} = (1 + \alpha g \ell)^{-\left(1 + \frac{N-2}{2\pi\alpha}\right)} \,. \tag{52}$$

Thus, the surface velocity flows to the bulk velocity as a power of logarithm of the length-scale.

For the case of fixed points at a finite small value of $g$, (e.g. the special fixed point for $N$ slightly below $N_c$ in scenario I, and the extra-ordinary-power fixed point for $N$ slightly above $N_c$ in scenario II), $v_s/v_b - 1$ flows to zero as a power of the length-scale.

## 5 Comparison with classical Monte Carlo.

In this section, we discuss the current state of Monte Carlo studies of boundary criticality in the classical $O(N)$ model in $d = 3$.

For $N = 2$ the phase diagram in Fig. 1 is well-established numerically and the critical exponents associated with the ordinary and the special universality classes have been extracted [19, 20, 22], see table 1. However, the nature of the extra-ordinary transition had remained unsettled until very recently (see below).

For $N = 3$ the ordinary boundary universality class is well studied with Monte Carlo [22], but the region of the phase diagram with $\kappa \gtrsim 1$ has received fairly little attention until very recently. The early study in Ref. [39] had concluded that while for $\kappa \leq 1.5$ the system crosses over to the ordinary universality class at large length scales, for $\kappa \geq 2$ the ordinary universality class is not reached for system sizes studied. A slightly later study, Ref. [22], had found a crossing point in the Binder ratio for the surface order parameter at $\kappa \approx 1.85$ that was taken to suggest the existence of a special fixed point. However, the properties of the putative extra-ordinary transition in the $\kappa > 1.85$ region were not studied in detail. A similar crossing point in the Binder ratio at $\kappa \approx 2.26$ was also observed for the $N = 4$ case in Ref. [40] and an attempt to extract critical exponents associated with the putative special transition has been made, see table 1.

After the arXiv version of the present paper came out, two further large scale Monte Carlo studies of the cases with $N = 2$ and $N = 3$ have appeared. [21,30] In Ref. [30] the $N = 3$ case was studied. The existence of a crossing point in the Binder ratio was confirmed and critical exponents associated with this special transition were extracted. Further, the behavior of the model for one value of $\kappa > \kappa_c$ was studied in some detail - the Monte Carlo results appear consistent with the extra-ordinary-log universality class. For instance, the spin-spin correlation function on the surface appears to be consistent with Eq. (2) with $q \approx 2.1(2)$, which translates to $\alpha \approx 0.15(2)$ via Eq. (38). Even more striking is the observed behavior of the spin-helicity modulus $\Upsilon$ defined as

$$\Upsilon = \frac{1}{L}\frac{\partial^2 F}{d\theta^2}\bigg|_{\theta=0}. \tag{53}$$

Here the system lives on a $L \times L_\parallel \times L_\parallel$ slab of thickness $L$ with surface dimensions $L_\parallel \times L_\parallel$. The boundary conditions along the surface directions are taken to be periodic-twisted, i.e a gauge flux $\theta$ in the $SO(2)$ subgroup of $SO(3)$ is inserted along one of the periodic surface directions. $F(\theta) = -\log Z$ is the free-energy with these boundary conditions. For a boundary conformal fixed point we expect $L\Upsilon$ to go to a constant. On the other hand, for our extra-ordinary-log fixed point, we expect $L\Upsilon$ to be dominated by the contribution from the stiffness of the surface order parameter $\vec{n}$,

$$L\Upsilon \approx \frac{2}{g(\ell)} \approx \frac{2}{g} + 2\alpha\log L, \tag{54}$$

where we used the RG flow in Eq. (36). (The factor of 2 appears because the slab has two surfaces). Ref. [30], indeed, finds that $L\Upsilon$ grows roughly logarithmically at the extra-ordinary transition with no sign of saturation for system sizes up to $L = L_\parallel = 384$ and estimates $\alpha \gtrsim 0.11$. This is roughly consistent with the value of $\alpha$ extracted from the two-point function of the order parameter.

Ref. [21] has very recently revisited the extra-ordinary transition in the $N = 2$ model. Their results are consistent with the extra-ordinary-log universality class. The two-point function of the surface order parameter at separation $x = L_\parallel/2$ is consistent with Eq. (2) with $q \approx 0.59(2)$, translating to $\alpha \approx 0.27(1)$. The spin-stiffness $L\Upsilon$ grows logarithmically with system size and gives $\alpha \approx 0.27(2)$. The value of $\alpha$ extracted appears to be independent of $\kappa$ for $\kappa > \kappa_c$, confirming the universality of the extra-ordinary-log transition.

Our RG analysis predicts that the value of $\alpha$ in the extra-ordinary-log phase is controlled by the universal OPE coefficients $a_\sigma$, $b_t$ at the normal fixed point via Eqs. (16), (33). The normal fixed point in the $O(N)$ model with $N = 2, 3$ was very recently studied by Monte Carlo in Ref. [41] and the constants $a_\sigma$, $b_t$ were extracted. We list the corresponding value of $\alpha$ as $\alpha_{norm}$ in Table 1. We see that $\alpha_{norm}$ agrees rather well with the value of $\alpha$ obtained from direct simulations of the extra-ordinary-log phase, listed in Table 1 as $\alpha_{eo}$.

Summarizing the results above: there is reasonable evidence for the existence of the extra-ordinary-log universality class in the $N = 2, 3$ models, suggesting that the critical value $N_c > 3$. The value of $\alpha$ decreases from $N = 2$ to $N = 3$ as expected. Further, the exponents $\Delta_{\vec{n}}^{spec}$ and $\nu_{spec}^{-1}$ decrease as $N$ increases from 1 to 4 and become very small at $N = 4$. This favors Scenario I in section 4 (Fig. 4, left). In fact, the smallness of these two exponents at $N = 4$ suggests that $N_c$ is close to 4. We further note that in Scenario I, if $N$ is sufficiently close to $N_c$ that the special fixed point $g_*^{spec}$ is perturbatively accessible, we expect the ratio $r_{spec} = \frac{(N-1)\nu_{spec}^{-1}}{4\pi\Delta_{\vec{n}}^{spec}}$ to approach the constant $\alpha$ of the extra-ordinary-log phase. Using the data in Table 1, $r_{spec}(N = 2) = 0.149(1)$ and $r_{spec}(N = 3) = 0.217(7)$. $r_{spec}(N = 2)$ is far from $\alpha(N = 2)$, so for $N = 2$ the special transition lies outside of the perturbative regime. On the other hand, $r_{spec}(N = 3)$ is reasonably close to the value $\alpha_{norm}(N = 3)$ in Table 1, suggesting that for $N = 3$ $g_*^{spec}$ is perturbatively accessible and again favoring Scenario I in section 4.

Table 1: Monte Carlo results for surface criticality in the classical $O(N)$ model in $d = 3$. $\Delta_{\vec{n}}^{ord}$ and $\Delta_{\vec{n}}^{spec}$ are respectively the scaling dimensions of the surface order parameter at the ordinary and special fixed points. $\nu_{spec}$ is the correlation length exponent at the special transition. $Q_4^{spec} = \frac{\langle \vec{m}_1^2 \rangle^2}{\langle (\vec{m}_1^2)^2 \rangle}$ is the surface Binder ratio at the special transition ($\vec{m}_1 = \sum_{i \in bound} \vec{S}_i$ is the surface magnetization). For $N \geq 2$, $\alpha_{eo}$ is the universal constant in the extra-ordinary-log phase, extracted from Eqs. (2), (38); $\alpha_{norm}$ is the value of $\alpha$ obtained from the constants $a_\sigma$, $b_t$ of the normal fixed point via Eqs. (16), (33).

| $N$ | $\Delta_{\vec{n}}^{ord}$ | $\Delta_{\vec{n}}^{spec}$ | $\nu_{spec}^{-1}$ | $Q_4^{spec}$ | $\alpha_{eo}$ | $\alpha_{norm}$ |
|---|---|---|---|---|---|---|
| 1 [22] | 1.2626 (15) | 0.364(1) | 0.715(1) | 0.626(1) | | |
| 2 [21, 22, 41] | 1.219 (2) | 0.325(1) | 0.608(4) | 0.840(1) | 0.27(1) | 0.300(5) |
| 3 [30, 41] | 1.187(2) | 0.264(1) | 0.36(1) | 0.9388(4) | 0.15(2) | 0.190(4) |
| 4 [40] | 0.9798(12) | 0.184(2) | 0.107(15) | 0.9825(8) | | |

# 6  $N = 2$. The role of vortices. Quantum models.

In our discussion of the $N = 2$ case, we have so far ignored the effect of vortices on the surface. We expect vortices to be irrelevant at the extra-ordinary-log fixed point described in section 4. Indeed, ignoring the coupling to the bulk, the scaling dimension of an $m$-fold vortex in $\vec{n}$ is $\Delta_{V^m} = \frac{\pi m^2}{g}$. The coupling $g$ flows to 0 in the IR, so we expect vortices to be highly irrelevant at the extra-ordinary-log fixed point. However, as we describe below, we expect that vortices do play a role at the special fixed point in the classical $O(2)$ model.

First, however, we note that we may also consider quantum models with $U(1)$ symmetry in 2+1D bulk dimensions. As a prototype consider the transition from a Mott insulator to a superfluid of bosons (e.g. in a Bose-Hubbard model). If the bulk transition is taking place at a constant (integer) boson density, it is described by the same 2+1D $O(2)$ model we've been considering up till now. However, the surface boson density at the transition need not match the bulk density. We may again model the boundary by the action: $S_{ordinary} + S_\varphi + S_{\varphi\phi}$ with

$$
\begin{aligned}
S_\varphi &= \frac{1}{2g} \int dx \, d\tau \left( (\partial_\tau \varphi)^2 + (\partial_x \varphi)^2 \right), \\
S_{\varphi\phi} &= -\frac{\tilde{s}}{2} \int dx \, d\tau \left( e^{i\varphi} \hat{\phi}^* + e^{-i\varphi} \hat{\phi} \right).
\end{aligned}
\tag{55}
$$

Here $e^{i\varphi} \sim n_1 + in_2$ is the boundary order parameter. The complex scalar $\hat{\phi}$ is the boundary operator corresponding to the bulk order parameter with the ordinary boundary condition.

Assume that there is a translational symmetry along the boundary with period $\delta$. Let the average excess boson number near the boundary over length $\delta$ be $\rho$. Just as in a purely 1+1D system, if $\rho$ is irrational, we expect that vortices in $e^{i\varphi}$ will be absent due to translational symmetry.[13] Likewise, if $\rho = \frac{p}{q}$ with $p$, $q$ - mutually prime integers, we expect only $q$-fold vortices $V^q$ of $e^{i\varphi}$ will be allowed.

Let us first analyze the phase diagram ignoring vortices. The scaling dimension $\Delta_{\hat{\phi}} \approx 1.219$

---

[13]Strictly speaking, the quantity controlling the quantum number of vortices under translation is not the excess density $\rho$, but $\rho - P$, where $P$ is the bulk polarization density. [42] However, for the simple Bose-Hubbard model on the square lattice $P = 0$. We thank Ashvin Vishwanath and Chong Wang for clarifying this point to us.

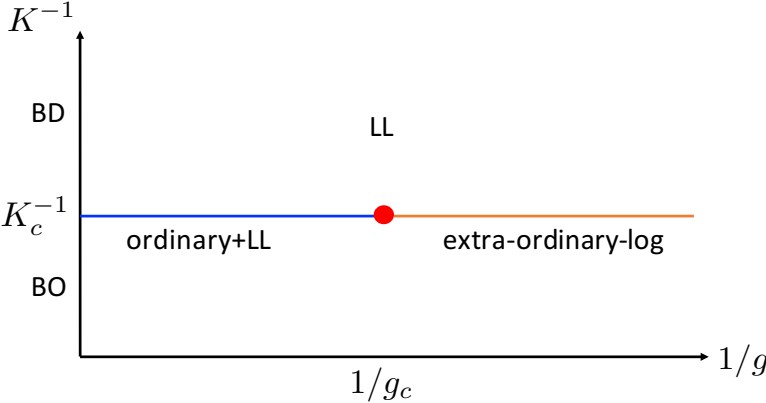

Figure 6: Conjectured phase diagram for the 2+1D Bose-Hubbard model with integer bulk filling and an incommensurate excess boson density $\rho$ on the boundary. $K$ is a non-thermal tuning parameter in the bulk and $g$ is the tuning parameter on the boundary. LL stands for bulk-disordered and boundary being a Luttinger-liquid. Ordinary+LL stands for a boundary Luttinger liquid essentially decoupled from the bulk with an ordinary boundary condition. In the case of commensurate excess boson density $\rho = \frac{p}{q}$ with $(p,q) = 1$ and $q \geq 3$, the phase diagram is essentially the same, except for the presence of an additional charge-density-wave boundary phase at large $g$.

for the ordinary universality class. [22] The scaling dimension $\Delta_{e^{i\varphi}} = \frac{g}{4\pi}$. Thus,

$$\frac{d\tilde{s}}{d\ell} = \left(2 - \Delta_{\hat{\phi}} - \frac{g}{4\pi}\right)\tilde{s}. \tag{56}$$

If $\frac{g}{4\pi} > g_c^0 = 2 - \Delta_{\hat{\phi}} \approx 0.781$, the coupling $\tilde{s}$ is irrelevant. In this regime, we have a bulk with ordinary boundary conditions with an effectively decoupled Luttinger liquid (LL) on the surface. We call this universality class LL+ordinary. On the other hand, for $g < g_c^0$, the coupling $\tilde{s}$ is relevant. One possibility is that the resulting flow is to the extra-ordinary-log fixed point at $g = 0$. Then we would have the phase diagram in Fig. 6. (Of course, we cannot rule out the existence of an additional stable boundary phase at intermediate values of $g$). Note that here we have two distinct stable boundary universality classes at $K = K_c$ that connect to the same phase for $K > K_c$.

Let us analyze the transition from the LL+ordinary to the extra-ordinary-log fixed point in more detail. We work perturbatively in $\tilde{s}$ and $g - g_c^0$. (The superscript zero on $g_c$ is to remind that this is the critical coupling when $\tilde{s} = 0$.) The structure of RG is very similar to that at the Kosterlitz-Thouless (KT) transition (and also to that discussed in Ref. [12]). We have the OPE:

$$\begin{aligned}
\hat{\phi}(\mathrm{x})\hat{\phi}^*(0) &\sim \frac{1}{\mathrm{x}^{2\Delta_\phi}}, \\
e^{i\varphi(\mathrm{x})}e^{-i\varphi(0)} &\sim \frac{1}{\mathrm{x}^{g/2\pi}}\left(1 - \frac{1}{4}\mathrm{x}^2(\partial_\mu\varphi)^2\right),
\end{aligned} \tag{57}$$

with $\mathrm{x} = (\tau, x)$. Here we have included only Lorentz scalars in the $e^{i\varphi(\mathrm{x})}$ OPE. Recalling that if

$$\delta S = -\lambda_i \int d^2\mathrm{x}\, O_i(\mathrm{x}), \tag{58}$$

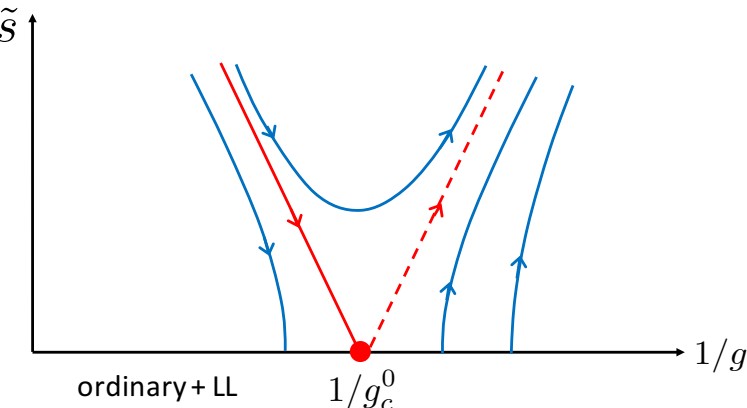

Figure 7: RG flow at the boundary of 2+1D Bose-Hubbard model in the absence of vortices. The solid red line marks a phase transition.

and $O_i(\mathbf{x})O_j(0) \sim \frac{C_{ijk}}{|\mathbf{x}|^2}O_k(0)$ then,

$$\frac{d\lambda_k}{d\ell} = \pi C_{ijk}\lambda_i\lambda_j\,, \tag{59}$$

we have

$$\frac{dg}{d\ell} = -\frac{\pi\tilde{s}^2}{4}g^2 \approx -4\pi^3(2-\Delta_{\hat\phi})^2\tilde{s}^2\,. \tag{60}$$

The RG equations (56), (60) are essentially the same as for a KT transition and result in the flow diagram in Fig. 7. Letting

$$u = \frac{g}{4\pi} - (2-\Delta_{\hat\phi})\,, \qquad v = \pi(2-\Delta_{\hat\phi})\tilde{s}\,, \tag{61}$$

we have

$$\begin{aligned}
\frac{dv}{d\ell} &= -uv\,,\\
\frac{du}{d\ell} &= -v^2\,.
\end{aligned} \tag{62}$$

We have the separatrix $u = v$ along which $u$ and $v$ flow to zero as $v(\ell) = \frac{v}{1+v\ell}$, and the attractive fixed line $u = -v$ along which $v(\ell) = \frac{v}{1-v\ell}$. If we start with initial $v$ and $u$ close to the separatrix $v = u$ with $v > u$ then the RG diverges at $\ell \approx \frac{\pi}{\sqrt{v^2-u^2}}$. This is the typical $\xi \sim \exp\left(const/\sqrt{g_c-g}\right)$ divergence of the correlation length characteristic of the KT transition. At the transition, $u = v$ flow to zero logarithmically and we have $\Delta_{e^{i\varphi}} = 2 - \Delta_{\hat\phi} \approx 0.781$.

In the present analysis we have ignored the possible difference between velocities on the surface and in the bulk. As we show in appendix B, taking the velocity difference into account does not qualitatively change the nature of the transition (even though the surface velocity does not flow to the bulk velocity at the transition.)

Another important assumption that we have made is that there are no $U(1)$ neutral relevant boundary operators at the ordinary fixed point. While we expect that this is so for Lorentz scalars (as the ordinary fixed point is stable), it is less obvious for Lorentz vectors, e.g. for the boundary operator corresponding to the bulk $U(1)$ current $j^\mu$ with $\mu = \tau, x$ along the surface. While such a vector is prohibited by e.g. rotational symmetry in the classical model, $\hat{j}^\tau$ is generally allowed in the quantum model. For the $O(N)$ model with $N \to \infty$, the $O(N)$

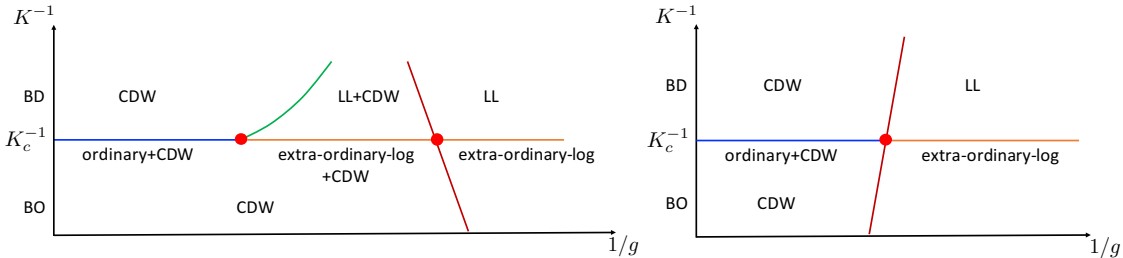

Figure 8: Possible phase diagrams for the 2+1D Bose-Hubbard model with integer bulk filling and a density $\rho = 1/2$ on the boundary. All phase labels refer to boundary behavior. While the phase diagram on the left is certainly possible, it is not clear if the one on the right can be realized with a direct continuous transition as a function of $g$ at $K = K_c$.

current $\hat{j}^{\tau,x}_{ab}$ has a boundary scaling dimension 3, so it is, indeed, irrelevant. Also, in $d = 4$, $\hat{j}^{\mu}$ has dimension $5 > d - 1$ for $\mu$ parallel to the surface.

Next, we analyze the effect of vortices on this transition. We have $\Delta_{V^m} = \frac{\pi m^2}{g_c^0} \approx 0.32 m^2$. Thus, vortices with $m = 1$ and $m = 2$ are relevant at the transition described above, while vortices with $m \geq 3$ are irrelevant. This implies that if the surface boson filling $\rho$ is irrational or if $\rho = \frac{p}{q}$ with $(p,q) = 1$ and $q \geq 3$ then all symmetry allowed vortices are irrelevant at $g = g_c^0$. On the other hand, if $\rho$ is an integer or a half-integer (e.g. if it is fixed to these values by a discrete symmetry) then there exist symmetry allowed vortices that are relevant at $g_c^0$. In the latter case vortices are also relevant for $g > g_c^0$, so not only is the transition unstable to vortices but also the LL+ordinary phase adjacent to it. For $q = 1$ one may expect that single vortices just destroy the Luttinger liquid leaving the ordinary universality class and also modifying the phase transition between the ordinary and the extra-ordinary-log phases. This then gives the same phase diagram as in the classical case, Fig. 1 (with $K$ and $K_1$ being non-thermal tuning parameters in the bulk and on the surface). For $q = 2$ we expect that for large $g$ double vortices drive the Luttinger liquid into a charge (or bond) density wave. There are then two possible scenarios for the evolution of the boundary as $g$ is decreased, see Fig. 8. In the more mundane scenario (Fig. 8, left), fixing $K = K_c$, as one decreases $g$ one first encounters a transition to the extra-ordinary-log universality class with co-existing charge-density-wave (CDW) order. This transition would be the same as in the classical $O(2)$ model. As one further decreases $g$ the charge-density-wave order disappears and we are left with pristine extra-ordinary-log universality class. There are certainly microscopic models which realize this mundane scenario. In the more interesting scenario (Fig. 8, right), one encounters just a single continuous transition as $g$ is decreased at which CDW disappears and the extra-ordinary-log behavior onsets. It is currently not clear if such a direct continuous transition is possible (a direct first order transition is, of course, not ruled out).

# 7  Quantum models: $N = 3$.

We begin this section by reviewing recent Monte Carlo results on quantum spin models in 2+1D with $SO(3)$ symmetry. [7–9, 43, 44] A prototypical Hamiltonian considered in Refs. [9, 43] is

$$H = \sum_{\langle ij \rangle} J_{ij} \vec{S}_i \cdot \vec{S}_j \, . \tag{63}$$

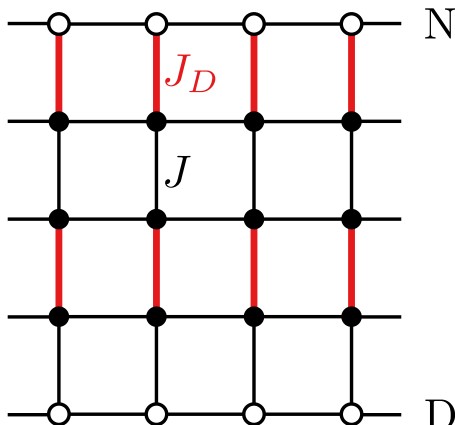

Figure 9: The quantum spin model considered in Refs. [9, 43] (figure taken from Ref. [43]). Red bonds have stronger coupling $J_D$ and black bonds have weaker coupling $J$. N and D mark edges with non-dangling and dangling spins respectively.

Here $\vec{S}_i$ is a spin-$S$ quantum spin on site $i$. The sites are arranged on a rectangular lattice, further, the couplings $J_{ij}$ are chosen to be dimerized, as shown in Fig. 9, with $J_D$ - the coupling on stronger (red) bonds and $J$ - the coupling on weaker (black) bonds. As one increases the ratio $K = J/J_D$ the bulk of the system goes from a trivial paramagnet to a Neel state. Numerical investigations confirm that this bulk transition is in the classical 3D $O(3)$ universality class. [45] However, unusual boundary behavior at this transition was found. In fact, two types of edges were investigated: the edge with "non-dangling" spins (Fig. 9, top edge) and the edge with "dangling" spins (Fig. 9, bottom edge). The boundary scaling dimension of the Neel order parameter $\Delta_{\vec{n}}$ was extracted: for the non-dangling edge it was found that $\Delta_{\vec{n}} \approx 1.15$ [9, 43], consistent with the ordinary universality class in the classical $O(3)$ model. However, for the dangling edge an exponent strikingly different from the ordinary universality class was found $\Delta_{\vec{n}} \approx 0.25$. [9, 43]

Initially, only models with $S = 1/2$ were considered. The unusual boundary behavior at the dangling edge was then attributed to the fact that when the bulk is in the paramagnetic phase, one effectively has a spin-1/2 Heisenberg chain on the boundary. Deep in the paramagnetic phase ($K \to 0$) this chain realizes the $SU(2)_1$ 1+1D CFT. If this CFT survives all the way to the bulk critical point, then it is natural that the boundary universality class at $K = K_c$ must be distinct from ordinary. However, somewhat surprisingly, a very similar exponent $\Delta_{\vec{n}}$ at $K = K_c$ was also found at the dangling edge of a model with $S = 1$, [43] where no gapless edge behavior is expected for $K < K_c$.[14] Further, the exponents found at the dangling edge of different microscopic models with $S = 1/2$ (e.g. with different lattice geometries) are numerically quite close. [7–9, 43] (Some drifts of exponents were, however, found in Ref. [43] when explicit perturbations to the boundary were considered.)

We now discuss theoretical expectations for boundary criticality at the dangling edge in the quantum model above. We may employ the same treatment as in the classical $O(3)$ model in sections 2, 3, 4. However, for quantum models we need to supplement the boundary action (4) with the topological $\theta$-term for the boundary Neel order parameter $\vec{n}$:

$$S_\theta = \frac{i\theta}{4\pi} \int dx d\tau\, \vec{n} \cdot (\partial_x \vec{n} \times \partial_\tau \vec{n}). \tag{64}$$

---

[14]In the parameter regime considered, the $S = 1$ model with $K < K_c$ realizes a trivial paramagnet, not a stack of Haldane chains. [45]

We expect that just as for the 1+1D chain, for the dangling edge geometry described above $\theta = 2\pi S$, i.e. $\theta = \pi$ for $S = 1/2$ and $\theta = 0$ (modulo $2\pi$) for $S = 1$.

Thus, for the $S = 1$ quantum model $\theta = 0$ and we expect all universal properties to be the same as for the classical model. As discussed in section 5, very recent Monte Carlo simulations [30] indicate the presence of stable ordinary and extra-ordinary-log fixed points in the classical $O(3)$ model and a special transition between them - we will rely on this interpretation of classical Monte Carlo results from here on. Next, for the $S = 1/2$ quantum model what is the effect of the $\theta = \pi$ term (64)? We recall that perturbation theory in $g$ is completely insensitive to the topological term (64). Indeed, $S_\theta$ is only non-zero for skyrmion configurations of $\vec{n}$, which are inaccessible in perturbation theory about the ordered state that we've employed in section 3. Thus, the perturbative $\beta$-function $\beta(g)$ is the same in the quantum $S = 1/2$ model as in the classical $O(3)$ model to all orders in $g$. Therefore, we expect the existence of a stable extra-ordinary-log phase in the $S = 1/2$ quantum model, with exactly the same universal parameter $\alpha$ as in the classical model governing the low-energy properties (2), (38), (54). Any differences between the $\theta = 0$ and $\theta = \pi$ cases will only enter through skyrmion configurations. Ignoring the coupling to the bulk, the classical skyrmion action is

$$S_{skyrm} = \frac{4\pi|m|}{g}, \tag{65}$$

where $m \in \mathbb{Z}$ is the skyrmion charge. Since $g$ logarithmically flows to zero in the extra-ordinary-log phase we expect skyrmion effects to be suppressed there. On the other hand, at the special fixed point in the classical model $g_*^{spec}$ is finite, therefore, we expect some differences in the critical properties between the $\theta = 0$ and $\theta = \pi$ cases at the special transition. However, given the relative smallness of the exponent $\Delta_{\vec{n}}^{spec} \approx 0.264(1)$ in the classical $O(3)$ model, [30] one may suspect that $g_*^{spec}$ is small to moderate. We then expect skyrmion effects to be suppressed at the special transition by $e^{-S_{skyrm}} = e^{-4\pi/g_*^{spec}}$, where we've replaced $g$ by its fixed point value. Using the relation (44), $e^{-S_{skyrm}} \approx e^{-\frac{2}{\Delta_{\vec{n}}^{spec}}} \approx e^{-8}$, where we've used the numerical value of $\Delta_{\vec{n}}^{spec}$ from Ref. [30] (of course, we don't know how large the prefactor of the exponential is). Thus, the exponents at the special transition might be numerically close in the $S = 1/2$ quantum and $O(3)$ classical models.

Finally, at the ordinary fixed point $g$ is large and perturbation theory in $g$ does not apply. We expect the $\theta = \pi$ term to play an important role here. One possibility is that the large $g$ phase at $\theta = \pi$ carries VBS (valence-bond-solid) boundary order, spontaneously breaking the translation symmetry along the edge. The ordinary phase in the $S = 1/2$ model is then described by the classical ordinary fixed point with an extra two-fold degeneracy of all states due to VBS order. Tuning slightly away from the bulk critical point into the paramagnetic bulk phase then leads to a VBS ordered dangling edge, consistent with the Lieb-Schultz-Mattis theorem. [46] We, thus, conjecture a boundary phase diagram for the $S = 1/2$ dangling edge in Fig. 10. This phase diagram was also put-forward in Ref. [12], which performed an RG study of the dangling edge starting from the $SU(2)_1$ $1 + 1$D CFT coupled to the ordinary boundary fixed point. (Strictly speaking, the nature of the small $g$ phase in Ref. [12] was left open, with the possibility of truly long-range boundary Neel order considered.)

What are the correlators of the VBS order parameter $V(x)$ at the extra-ordinary-log and special fixed points? First, $V(x)$ can be obtained from the dimer operator $\vec{S}_i \cdot \vec{S}_{i+1} \sim const + (-1)^i V(x)$, where $i$ is the coordinate along the boundary. Under translations, $T_x : \vec{n} \to -\vec{n}$, $V \to -V$. Based on symmetries, we identify $V$ with the skyrmion density,

$$V(x) \sim i\, \vec{n} \cdot (\partial_x \vec{n} \times \partial_\tau \vec{n}). \tag{66}$$

This identification holds for both $S = 1/2$ and $S = 1$ models. We expect that perturbatively in $g$, $V(x)$ does not receive any anomalous dimensions. Indeed, such an anomalous dimension

would result in a flow of $\theta$, which would spoil the periodicity of $\theta$. Thus, we expect that up to non-perturbative skyrmion effects, the boundary scaling dimension $\Delta_V = 2$. Since skyrmions are suppressed in the extra-ordinary-log phase, we expect $\Delta_V = 2$ there. If the special fixed point occurs at weak coupling then skyrmions are also partially suppressed there and $\Delta_V^{spec} \approx 2$.

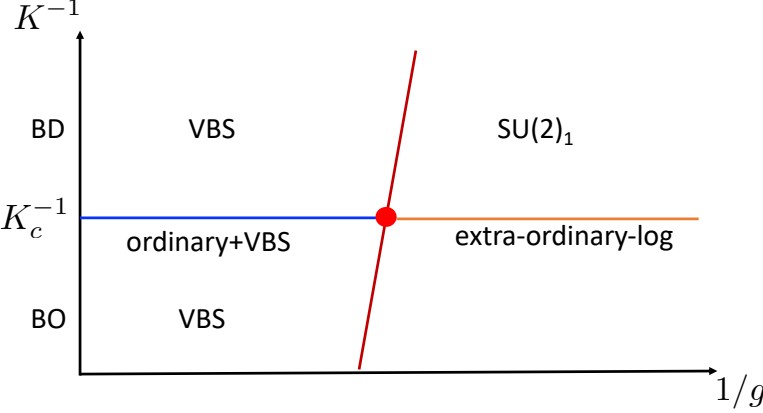

Figure 10: A possible phase diagram for the 2+1D spin $S = 1/2$ model with a "dangling" edge.

How does the above theoretical picture compare to Monte Carlo studies of quantum spin models? We recall that quantum Monte Carlo simulations for both $S = 1/2$ and $S = 1$ models find power-law behavior of the two-point function of the Neel order parameter on the dangling edge with $\Delta_{\vec{n}} \approx 0.25$. This value is quite close to $\Delta_{\vec{n}}^{spec} = 0.264(1)$ found in the classical $O(3)$ model at the special transition. [30] Thus, perhaps the dangling edge in quantum models of Refs. [7–9, 43] is controlled by the special fixed point. This possibility was first put forward in Ref. [8], which pointed out that $\Delta_{\vec{n}}$ at the dangling edge is quite close to the value of $\Delta_{\vec{n}}^{spec}$ obtained using the $4 - \epsilon$ expansion for the boundary special fixed point in the classical $O(3)$ Wilson-Fisher model. (However, the edge phase diagram of the classical $O(3)$ model in 3d was not understood at the time). Ref. [12] also suggested that the dangling edge in the Monte Carlo simulations of $S = 1/2$ models is controlled by the special fixed point in Fig. 10, i.e. that the quantum models that have been numerically studied so far are *accidentally* tuned close to the special transition. This interpretation is slightly surprising, given that a number of different quantum models with different geometries and both $S = 1/2$ and $S = 1$ have been studied. [7–9, 43, 44] One point in favor of this interpretation is that the correlation length exponent $\nu_{spec}^{-1} \approx 0.36(1)$ found by Monte Carlo in the classical $O(3)$ model is quite small, so that if a microscopic model is accidentally tuned to the vicinity of the special transition, the length-scale $\xi \sim |g - g_c|^{-\nu}$ for the cross-over to the true IR behavior (either the ordinary or the extra-ordinary-log phase) is enhanced. The slight drift in $\Delta_{\vec{n}}$ at the dangling edge observed in Ref. [43] as the edge coupling was varied might also be consistent with this interpretation. Further, as already pointed out, the observed similarity in $\Delta_{\vec{n}}$ in the $S = 1/2$ and $S = 1$ models [43] might be due to the smallness of $g_*^{spec}$ and the consequent partial suppression of skyrmions at the special fixed point. Another piece of the puzzle is the recent study in Ref. [47] of the VBS two-point function at the dangling edge. Although a precise scaling dimension $\Delta_V$ is difficult to extract, for the $S = 1/2$ model the authors estimate $\Delta_V \approx 1.2 - 1.4$. For $S = 1$ the two point function of $V$ falls off very quickly making a reliable estimate of $\Delta_V$ even more difficult, however, the data appears consistent with $\Delta_V = 2$ in this case. As discussed above, if $g_*^{spec}$ is small, up to non-perturbative corrections in $g_*^{spec}$, we expect $\Delta_V^{spec} = 2$ which is at odds with the Monte Carlo result for the $S = 1/2$ model. It is not clear why non-perturbative

skyrmion effects at the special transition would be more pronounced in the VBS correlators and not in Neel correlators.

An alternative interpretation of the quantum Monte Carlo data initially put forward in the first arXiv version of the present paper is that in the classical $O(N)$ model scenario II in Fig. 3 is realized and, further, that $N_c < 3 < N_{c2}$. The classical $O(3)$ model would then possess an extra-ordinary-power phase, which could control the dangling edge in the quantum models studied by Monte Carlo. This would avoid the assumption of accidental fine-tuning to the vicinity of the special transition in the quantum models. The fact that the values of $\Delta_{\vec{n}}$ seen in the $S = 1/2$ and $S = 1$ models are close to each other could again be explained by the smallness of $g_*^{ext-p}$ at the extra-ordinary-power fixed point and the partial suppression of skyrmions. However, such an interpretation appears at odds with the observation of the extra-ordinary-log phase in the classical $O(3)$ model for large $\kappa$ (and, in particular, absence of saturation in the stiffness $L\Upsilon$ for large system sizes).

Very recently, the phase diagram of the non-dangling edge as a function of the edge spin-spin coupling was investigated with quantum Monte Carlo. [48] Theoretically, we would expect this phase diagram to be identical to that in the classical $O(3)$ model. However, Ref. [48] reports a special transition with exponents distinct from those found by Ref. [30] for the special transition in the classical $O(3)$ model. Ref. [48] also finds an extra-ordinary phase which does not appear to be consistent with extra-ordinary-log behavior; rather, the authors suggest that the extra-ordinary phase here supports true long range order. We don't currently have an interpretation of these results.

# 8 Discussion.

In this paper we have re-examined the boundary critical behavior of the $O(N)$ model in $d = 3$. We have established the phase diagram in the limit of $N$ close to 2 and for large but finite $N$, and have discussed two scenarios for the evolution of the system between these two limits, see Figs. 3, 4. Some important questions left unanswered by this work are: i) what is the critical value $N_c$ at which the extra-ordinary-log fixed point disappears; ii) which of the two scenarios in Figs. 3, 4 is realized. As we have discussed, the value of $N_c$ is determined by the universal amplitudes $a_\sigma(N)$ and $b_t(N)$ at the normal fixed point—we comment below on very recent numerical and analytical progress in computing these amplitudes. As for the question of which of the two scenarios in Figs. 3, 4 is realized, as we have discussed, this is determined by the sign of the coefficient $b(N_c)$ in the $\beta$-function (43). We expect that one of the inputs into $b$ is the boundary four-point function of $\hat{t}$ at the normal transition. It would be interesting to compute $b(N)$ for large $N$ and attempt to extrapolate to $N = N_c$. We leave this study to future work.

After the first version of this paper came out on arXiv, two large scale Monte Carlo studies of the classical $O(N)$ model with $N = 3$ [30] and $N = 2$ [21] have appeared. The results of these studies are consistent with our theoretical findings. In particular, behavior compatible with the extra-ordinary-log phase is found in the large $\kappa$ region of models with $N = 2$ [21] and $N = 3$ [30]. Further a special transition is observed not only for $N = 2$, but also for $N = 3$. [22, 30] These findings imply $N_c > 3$. The trend of critical exponents at the special transition as $N$ is increased for $N = 2, 3$ [21, 30] and earlier results for $N = 4$ [40] tentatively suggest that the first scenario in Figs. 3, 4 is realized. Further, Ref. [41] studied the normal universality class with Monte Carlo for $N = 2, 3$ and extracted the values of the universal coefficients $a_\sigma$, $b_t$: the results are in good agreement with our predictions for the relation between the normal and extra-ordinary-log fixed points.

Very recently the normal universality class of the $O(N)$ model with $N \geq 2$ was studied using

numerical conformal bootstrap. [33] Two methods were employed: the truncated bootstrap and the positive bootstrap. Estimates of $a_\sigma$ and $b_t$ were obtained with truncated bootstrap, which indicate $N_c \approx 5$. In addition, using the positive bootstrap, under a certain set of assumptions rigorous bounds were placed on $a_\sigma$ and $b_t$, which lead to a bound $N_c > 3$ (conditions under which a stronger bound $N_c > 4$ can be obtained were also discussed.)

The present work was largely motivated by Monte Carlo studies of boundary critical behavior in 2+1D quantum spin models with $SO(3)$ symmetry. [7–9,43,44] As we have discussed in section 7, boundary behavior distinct from the ordinary class is observed at the dangling edge in a number of models. We have discussed an interpretation where the dangling edge in models studied is accidentally tuned close to the special transition. [8,12] To test this interpretation, one should study larger system sizes, where an eventual cross-over to the extra-ordinary-log or the ordinary boundary phase is expected. A more extensive study of the stability of the observed behavior to edge perturbations, which could tune the system away from the special transition, would also be valuable.

Above all, we hope that the present work will lead to more detailed numerical studies of boundary critical behavior in both classical and quantum models with $SO(N)$ symmetry. It might also be possible to numerically study the behavior in these models as a continuous function of $N$ by reformulating them as loop models. [25]

# Acknowledgements

I am grateful to Ashvin Vishwanath for pointing out the numerical results in Refs. [7–9,43] to me, as well as for collaboration on related projects. I would also like to thank Shumpei Iino, Chao-Ming Jian, Zohar Komargodski, Jian-Ping Lv, Slava Rychkov, T. Senthil, Chong Wang, Lukas Weber, Stefan Wessel and Cenke Xu for discussions. I am particularly grateful to Hans Werner Diehl for his comments and pointers to the existing literature on the subject. I am very grateful to Ilya Gruzberg, Abijith Krishnan, Marco Meineri, Jay Padayasi, as well as Francesco Parisen Toldin, for many discussions and collaborations on related projects. This work is supported by the National Science Foundation under grant number DMR-1847861.

# A    The normal universality class in the large-$N$ expansion.

In this appendix, we derive Eq. (17) for the universal amplitudes characterizing the normal boundary of the $O(N)$ model in the large-$N$ limit. We follow Ref. [35].

We begin with

$$L = \frac{1}{2} \sum_{a=1}^{N} (\partial_\mu \phi_a)^2 + \frac{i\lambda}{2} \left( \sum_{a=1}^{N} \phi_a^2 - \frac{1}{g_{bulk}} \right). \tag{67}$$

We will work in $d = 3$ and place the boundary at $z = 0$. The coupling $g_{bulk}$ is assumed to be tuned to the critical point.

At $N = \infty$ we look for the saddle point,

$$\langle \phi_N(z) \rangle = \sigma_0(z) = \frac{a_\sigma^0}{(2z)^{1/2}},$$

$$\langle i\lambda(z) \rangle = i\lambda_0(z) = \frac{3}{4z^2}. \tag{68}$$

The coefficient of $\lambda_0$ is chosen so that $\sigma_0$ satisfies the saddle-point equation

$$(-\partial^2 + i\lambda_0(z))\sigma_0(z) = 0. \tag{69}$$

Note that at this point we are using a $\phi$ field that is not normalized. We will fix the normalization later.

We define the $\phi$ propagator

$$\langle \phi^i(x)\phi^j(x')\rangle = \delta^{ij}G(x,x') = \frac{\delta^{ij}}{(4zz')^{\Delta_\phi}}g(v), \qquad v = \frac{z^2+z'^2+\rho^2}{2zz'}, \quad \rho = |\mathbf{x}-\mathbf{x}'|, \quad (70)$$

where here and below $i,j = 1\ldots N-1$. We denote $G$ at $N = \infty$ by $G^0$ (similarly for $g$, $g^0$). We have

$$\mathcal{L}_x G^0(x,x') = \delta^3(x-x'), \qquad \mathcal{L}_x = \left(-\partial_x^2 + \frac{3}{4z^2}\right). \quad (71)$$

A useful identity is

$$\mathcal{L}_x \frac{1}{\sqrt{zz'}}p(v) = -\frac{1}{z^2}\frac{1}{\sqrt{zz'}}(\mathcal{D}p)(v), \quad (72)$$

with $v$ as in Eq. (70), $p(v)$ - an arbitrary function, and

$$(\mathcal{D}p)(v) = (v^2-1)p''(v) + 3vp'(v). \quad (73)$$

Thus, $\mathcal{D}g^0(v) = 0$ away from the singularity at $v = 1$. Solving for $g^0$ then gives $g^0(v) = c_1 \frac{v}{\sqrt{v^2-1}} + c_2$. The constant $c_1$ can be obtained by matching to the singular behavior of the bulk propagator $G^0_{bulk}(x,x') = \frac{1}{4\pi|x-x'|}$ as $x \to x'$, while $c_2$ is obtained by demanding that $g^0(v) \to 0$ as $v \to \infty$ (clustering). Then

$$g^0(v) = \frac{1}{2\pi}\left(\frac{v}{\sqrt{v^2-1}} - 1\right), \quad (74)$$

from which $g^0(v) \to \frac{1}{4\pi v^2}$ as $v \to \infty$, i.e.

$$(b_t^0)^2 = \frac{1}{16\pi}. \quad (75)$$

Again, this is without taking the normalization of $\phi$ into account.

Finally, the amplitude $a_\sigma^0$ is obtained from

$$\sum_{a=1}^N \langle \phi^a(x)\phi^a(x)\rangle = \frac{1}{g_c}. \quad (76)$$

Ignoring the fluctuations of $\phi_N$ (which only contribute an $O(1)$ term to the LHS),

$$(N-1)G^0(x,x) + \sigma_0^2 = \frac{1}{g_{bulk}}. \quad (77)$$

We can regularize $G^0(x,x)$ by taking the coincident limit of $G^0(x,x')$ as $x \to x'$, $G^0(x,x') \to \frac{1}{4\pi}\left(\frac{1}{s} - \frac{1}{z}\right)$, $s = |x-x'|$. Fixing a finite $s$, we get

$$(a_\sigma^0)^2 = \frac{N-1}{2\pi}. \quad (78)$$

Taking the bulk normalization of $\phi$ into account, we have agreement with the values of $a_\sigma^2$ and $b_t^2$ in (17) to leading order in $N$.

We wish to compute the $1/N$ correction to $G$ in order to extract the $1/N$ correction to $b_t$. We note that Eq. (76) is actually exact by equation of motion for $\lambda$. Let's define the connected "longitudinal" correlation function

$$G_\sigma(x,x') = \langle \phi_N(x)\phi_N(x')\rangle - \langle \phi_N(x)\rangle\langle \phi_N(x')\rangle \quad (79)$$

and the mixed correlation function

$$G_m(x, x') = \frac{1}{N}((N-1)G(x, x') + G_\sigma(x, x')).$$
(80)

Then Eq. (76) becomes

$$NG_m(x, x) + \langle \phi_N(x) \rangle^2 = \frac{1}{g_{bulk}}.$$
(81)

Thus, $\langle \phi_N \rangle$ can be extracted from the short-distance behavior of $G_m$. The same short-distance behavior determines the bulk normalization of the field $\phi^a(x)$, since in the absence of a boundary there is no difference between longitudinal and transverse components of $\phi^a$. Finally, the behavior of $G_m(x, x')$ for $\rho \to \infty$ (with fixed $z, z'$) is dominated by the term involving $G(x, x')$. Indeed, recall $\Delta_{\hat{D}} = 3$, so $G_\sigma(x, x')$ decays faster than $G(x, x')$ for $\rho \to \infty$. Thus, we can also extract $b_t$ from $G_m$. So we will focus on computing $1/N$ corrections to $G_m$ below.

In order to proceed, we need the propagator for $\lambda$. Let $\lambda = \lambda_0 + \delta\lambda$ and $\phi_N = \sigma_0 + \delta\sigma$. The action then becomes:

$$\begin{aligned}
L = &\frac{1}{2}(\partial_\mu \phi^i)^2 + \frac{1}{2}i\lambda_0(\phi^i)^2 + \frac{1}{2}(\partial_\mu \delta\sigma)^2 + \frac{1}{2}i\lambda_0\delta\sigma^2 \\
&+ i\sigma_0\delta\lambda\delta\sigma + \frac{1}{2}i\delta\lambda\left((\phi^i)^2 - \frac{1}{g_c}\right) + \frac{1}{2}i\delta\lambda\delta\sigma^2.
\end{aligned}$$
(82)

Integrating $\phi$ out, we obtain the following action for $\delta\lambda, \delta\sigma$ to quadratic order:

$$S_2[\delta\lambda, \delta\sigma] = \frac{1}{2}\int d^3x d^3x' \begin{pmatrix} \delta\sigma(x) \\ \delta\lambda(x) \end{pmatrix}^T \begin{pmatrix} \mathcal{L}_x\delta(x-x') & i\sigma_0(x)\delta(x-x') \\ i\sigma_0(x)\delta(x-x') & \frac{N-1}{2}G^0(x, x')^2 \end{pmatrix} \begin{pmatrix} \delta\sigma(x') \\ \delta\lambda(x') \end{pmatrix}.$$

Further integrating out $\delta\sigma$,

$$S_2[\delta\lambda] = \frac{1}{2}\frac{N-1}{2}\int d^3x d^3x' \delta\lambda(x)\Pi(x, x')\delta\lambda(x'),$$
(83)

with

$$\Pi(x, x') = G^0(x, x')^2 + \frac{2}{N-1}\sigma_0(x)G^0(x, x')\sigma_0(x').$$
(84)

Thus, the $\lambda$ propagator defined as

$$D^0_{\lambda\lambda}(x, x') = \langle \delta\lambda(x)\delta\lambda(x') \rangle, \quad N \to \infty,$$
(85)

satisfies $\frac{N-1}{2}\int d^3x' D^0_{\lambda\lambda}(x, x')\Pi(x', y) = \delta^3(x-y)$. We direct the reader to Ref. [35] for the details of how to compute $D^0_{\lambda\lambda}$. Here we just cite the result[15]:

$$D^0_{\lambda\lambda}(x, x') = -\frac{16}{(N-1)\pi^2 z^2 z'^2}\frac{v}{(v^2-1)^2} = \frac{16}{(N-1)\pi^2}\left(\frac{1}{((z+z')^2+\rho^2)^2} - \frac{1}{((z-z')^2+\rho^2)^2}\right).$$
(86)

Note that $D^0_{\lambda\lambda}$ has the correct form dictated by conformal invariance for a scalar of dimension $\Delta_\lambda = 2$. The other propagators can be expressed in terms of $D^0_{\lambda\lambda}$ and $G^0$:

$$\begin{aligned}
G^0_\sigma(x, x') &= \langle \delta\sigma(x)\delta\sigma(x') \rangle \\
&= G^0(x, x') - \int d^3y d^3y' G^0(x, y)\sigma_0(y)D^0_{\lambda\lambda}(y, y')\sigma_0(y')G_0(y', x'), \\
D^0_{\lambda\sigma}(x, x') &= \langle \delta\lambda(x)\delta\sigma(x') \rangle = -\int d^3y D^0_{\lambda\lambda}(x, y)i\sigma_0(y)G^0(y, x').
\end{aligned}$$
(87)

---

[15]The associated Legendre functions in Eq. 3.22 of Ref. [35] simplify for $d = 3$.

We are now ready to compute $1/N$ corrections to $G_m$. First, we let $G^{-1}(x,x') = G^{0^{-1}}(x,x') + \Sigma(x,x')$ and $G_m^{-1}(x,x') = G^{0^{-1}}(x,x') + \Sigma_m(x,x')$. We then have to leading order in $1/N$:

$$\Sigma(x,x') = \Sigma^a(x,x') + \Sigma^b(x,x') + \Sigma^c(x,x'), \qquad (88)$$

with diagrams for each term shown in Fig. 11. Explicitly,

$$
\begin{aligned}
\Sigma^a(x,x') &= D^0_{\lambda\lambda}(x,x')G^0(x,x'), \\
\Sigma^b(x,x') &= \Sigma^b_1(x,x') + \Sigma^b_2(x,x'), \\
\Sigma^b_1(x,x') &= -\frac{N-1}{2}\delta^3(x-x')\int d^3y\,d^3w\,d^3w'\,D^0_{\lambda\lambda}(x,y)G^0(y,w)D^0_{\lambda\lambda}(w,w')G^0(w,w')G^0(w',y), \\
\Sigma^b_2(x,x') &= \frac{1}{2}\delta^3(x-x')\int d^3y\,D^0_{\lambda\lambda}(x,y)G^0_\sigma(y,y), \\
\Sigma^c(x,x') &= \delta^3(x-x')\int d^3y\,D^0_{\lambda\sigma}(x,y)D^0_{\lambda\sigma}(y,y).
\end{aligned}
\qquad (89)
$$

Further, since $G_\sigma$ appears in Eq. (80) with a factor of $1/N$, we may replace it by $G^0_\sigma$ in computing $G_m$ to order $1/N$. We then have to $O(1/N)$,

$$\Sigma_m(x,x') = \Sigma^a_m(x,x') + \Sigma^b_m(x,x') + \Sigma^c_m(x,x'), \qquad (90)$$

with

$$
\begin{aligned}
\Sigma^a_m(x,x') &= \frac{N-1}{N}\left(D^0_{\lambda\lambda}(x,x')G^0(x,x') + \frac{1}{N-1}\sigma_0(x)D^0_{\lambda\lambda}(x,x')\sigma_0(x')\right), \\
\Sigma^b_m(x,x') &= \frac{N-1}{2N}\delta^3(x-x')\int d^3y\,D^0_{\lambda\lambda}(x,y)G^0(y,y) \\
&\quad - \frac{N-1}{2}\delta^3(x-x')\int d^3y\,d^3w\,d^3w'\,D^0_{\lambda\lambda}(x,y)G^0(y,w)\Sigma^a_m(w,w')G^0(w',y), \\
\Sigma^c_m(x,x') &= -\delta^3(x-x')\int d^3y\,d^3w\,d^3w'\,D^0_{\lambda\lambda}(x,w)\sigma_0(w)G^0(w,y)\Sigma^a_m(y,w')\sigma_0(w') \qquad (91)\\
&\quad + \frac{1}{N}\delta^3(x-x')\int d^3y\,d^3w\,d^3w'\,D^0_{\lambda\lambda}(x,w)\sigma_0(w)G^0(w,y)\sigma_0(y)D^0_{\lambda\lambda}(y,w')\sigma^2_0(w').
\end{aligned}
$$

(Strictly speaking, to the accuracy we are working, we should set factors of $(N-1)/N$ to 1 in the above equations. However, it is interesting to observe that what seemed like an expansion in $(N-1)^{-1}$, to this order appears to organize itself as an expansion in $N^{-1}$.)

We now observe that the last term in equation for $\Sigma^c_m$ vanishes, as does the first term in equation for $\Sigma^b_m$ (apart for a shift of the critical value of $g_{bulk}$.) Indeed, going to mixed position-momentum space, we have

$$D^0_{\lambda\lambda}(z,z',p=0) = \int d^2\mathrm{x}\,D^0_{\lambda\lambda}(\mathrm{x},z;0,z') = \frac{16}{(N-1)\pi}\left(\frac{1}{(z+z')^2} - \frac{1}{(z-z')^2}\right).$$

How to treat the singularity at $z = z'$ in the last term above? This singularity comes from Fourier transforming the last term in Eq. (86), which is nothing but the bulk $\lambda$ propagator. Indeed, the singular behavior of $D^0_{\lambda\lambda}(x,x')$ as $x \to x'$ is the same as in the absence of a boundary: if we consider the OPE, $\delta\lambda(x)\delta\lambda(0) \sim \frac{N_\lambda}{x^{2\Delta_\lambda}} + C_{\lambda\lambda\lambda}\frac{1}{x^{\Delta_\lambda}}\delta\lambda(0) + \ldots$, with $N_\lambda \approx -\frac{16}{N\pi^2}$ then the subleading term $C_{\lambda\lambda\lambda} \sim \frac{1}{N^3}$ does not contribute at $N = \infty$ [49]. Now, in the absence of a boundary we know

$$D_{\lambda\lambda}(q) = \int d^3x\,D^0_{\lambda\lambda}(x,0)e^{-iqx} = \frac{16}{N-1}q, \qquad (92)$$

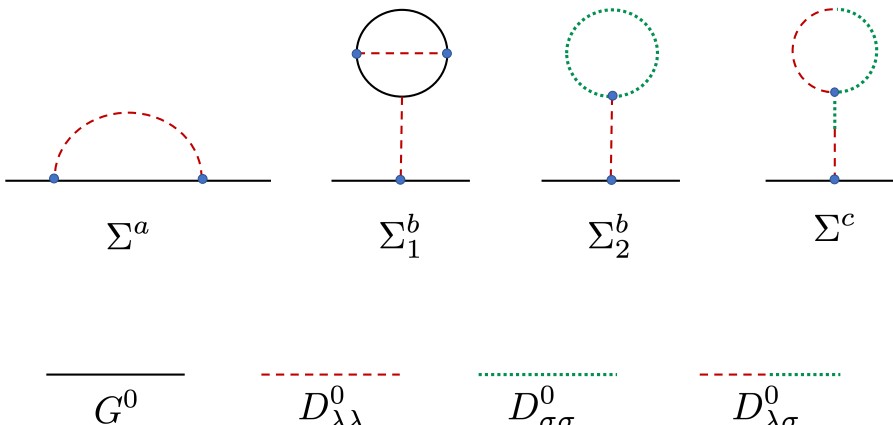

Figure 11: Normal fixed point of the $O(N)$ model. Diagrams contributing to the self-energy $\Sigma(x, x')$, Eq. (89), at $O(1/N)$. Notation for propagators is introduced in the bottom of the figure; blue dots mark interaction vertices.

which after Fourier transforming gives

$$D_{\lambda\lambda}(z, z'; p = 0) = \frac{16}{(N-1)\pi} \frac{d}{dz} \frac{P}{z - z'}, \tag{93}$$

with $P$ denoting principal value. Thus, in the presence of a boundary,

$$D^0_{\lambda\lambda}(z, z', p = 0) = \frac{16}{(N-1)\pi} \frac{d}{dz} \left( \frac{P}{z - z'} - \frac{1}{z + z'} \right). \tag{94}$$

Now it is easy to check that

$$\int_0^\infty \frac{dz'}{z'} D^0_{\lambda\lambda}(z, z', p = 0) = 0. \tag{95}$$

Thus, the integral over $w'$ in the last term of $\Sigma^c_m$ in Eq. (91) vanishes. Likewise, in the first term in $\Sigma^b_m$, $G^0(y, y) = \frac{1}{4\pi}(\frac{1}{\epsilon} - \frac{1}{y_d})$, where $\epsilon$ is the UV cut-off. The $\epsilon^{-1}$ term shifts the location of $g_{bulk,c}$, while the contribution of the $y_d^{-1}$ term vanishes upon integrating over $y$.

Thus, defining $G^a_m$ as the contribution of $\Sigma^a_m$ to $G_m$, i.e. $G^a_m(x, x') = -\int d^3y\, d^3y'\, G^0(x, y) \Sigma^a_m(y, y') G^0(y', x')$, we have

$$\Sigma^b_m(x, x') = \frac{N-1}{2} \delta^3(x - x') \int d^3y\, D^0_{\lambda\lambda}(x, y) G^a_m(y, y),$$

$$\Sigma^c_m(x, x') = \delta^3(x - x') \int d^3w\, d^3w'\, D^0_{\lambda\lambda}(x, w) \sigma_0(w) \left[ \mathcal{L}_{w'} G^a_m(w, w') \right] \sigma_0(w'). \tag{96}$$

Next, we compute $G^a_m$. We have

$$\Sigma^a_m(x, x') = -\frac{4}{N\pi^3} \frac{1}{(zz')^{5/2}} \frac{v^2}{(v^2 - 1)^{5/2}}. \tag{97}$$

If not for the UV divergence of $\Sigma^a_m(x, x')$ as $x \to x'$, $G^a_m$ would transform under conformal transformations as a two-point function of a scalar of dimension 1/2. Let's study how the cut-off dependence modifies this. We regularize,

$$G^a_m(x, x') = -\int_{|y - y'| > a} d^3y\, d^3y'\, G^0(x, y) \Sigma^a_m(y, y') G^0(y', x'), \tag{98}$$

where $a$ is a short-distance cut-off. Consider an infinitesimal conformal transformation $x^\mu \to \zeta^\mu(x) \approx x^\mu + \epsilon^\mu(x)$. Then

$$\frac{\partial \zeta^\mu}{\partial x^\rho} \frac{\partial \zeta^\mu}{\partial x^\sigma} = \Omega^2(x)\delta_{\rho\sigma}, \tag{99}$$

$\Omega(x) \approx 1 + \frac{1}{d}\partial_\rho \epsilon^\rho$. For a scalar primary $O(x)$ of dimension $\Delta$,

$$\langle O(x)O(y)\rangle = \Omega(x)^{\Delta_O}\Omega(y)^{\Delta_O}\langle O(\zeta(x))O(\zeta(y))\rangle. \tag{100}$$

Noting that $\Sigma_m^a$ has the form of a two-point function of a conformal scalar of dimension $5/2$, Eq. (13), we have

$$G_m^a(x,x') = -\Omega(x)^{1/2}\Omega(x')^{1/2}\int_{|y-y'|>a} d^3y\, d^3y'\, \Omega^3(y)\Omega^3(y') \tag{101}$$

$$\times G^0(\zeta(x),\zeta(y))\Sigma_m^a(\zeta(y),\zeta(y'))G^0(\zeta(y'),\zeta(x')).$$

Now changing variables in the integral,

$$\delta_\epsilon G_m^a \equiv G_m^a(x,x') - \Omega(x)^{1/2}\Omega(x')^{1/2}G_m^a(\zeta(x),\zeta(x'))$$

$$\approx -\int d^3y\, d^3y'\, G^0(x,y)\Sigma_m^a(y,y')G^0(y',x') \tag{102}$$

$$\times \left(\theta(|\zeta^{-1}(y)-\zeta^{-1}(y')|-a)-\theta(|y-y'|-a)\right),$$

where we have only kept terms to first order in $\epsilon$. Expanding the difference of $\theta$ functions,

$$\delta_\epsilon G_m^a \approx \int d^3y\, d^3y'\, \delta(|y-y'|-a)\frac{(y-y')\cdot(\epsilon(y)-\epsilon(y'))}{|y-y'|}G^0(x,y)\Sigma_m^a(y,y')G^0(y',x'). \tag{103}$$

We now expand the integrand in $s = y' - y$. We have:

$$\Sigma_m^a(y,y') = -\frac{4}{N\pi^3}\left(\frac{1}{s^5} + \frac{3}{8y_d^2}\frac{1}{s^3} + \ldots\right), \tag{104}$$

$G^0(y',x) = (1 + s^\mu\partial_\mu^y + \frac{1}{2}s^\mu s^\nu\partial_\mu^y\partial_\nu^y + \ldots)G^0(y,x)$. There are two types of conformal transformations that we need to consider: scale transformations, $\epsilon^\mu(x) = \epsilon x^\mu$, and special conformal transformations, $\epsilon^\mu(x) = b^\mu x^2 - 2(b\cdot x)x^\mu$, with $b$ - entirely in the boundary plane ($b_z = 0$). Let's begin with scale transformations. Performing the integral over $s^\mu$ (and keeping only finite terms in $a$), we have

$$\delta_\epsilon G_m^a(x,x') = -\frac{16\epsilon}{N\pi^2}\int d^3y\, G^0(x,y)\left(\frac{1}{6}\partial_y^2 + \frac{3}{8}\frac{1}{y_d^2}\right)G^0(y,x')$$

$$= \frac{8\epsilon}{3N\pi^2}G^0(x,x') - \frac{8\epsilon}{N\pi^2}\int d^3y\, G^0(x,y)\frac{1}{y_d^2}G^0(y,x'). \tag{105}$$

In the last step we have used $\mathcal{L}_y G^0(y,x') = \delta^3(y-x')$. Note that we have dropped a term which diverges as $a^{-2}$. This term will be cancelled by a shift in the expectation value, $\langle\delta\lambda\rangle$, at the critical point. Similarly, under special conformal transformations,

$$\delta_\epsilon G_m^a(x,x') = \frac{4}{N\pi^3}\int d^3y\, d^3s\, G^0(x,y)\delta(s-a)s(2b\cdot y + b\cdot s)\left(\frac{1}{s^5} + \frac{3}{8y_d^2}\frac{1}{s^3}\right)$$

$$\times \left(1 + s^\mu\partial_\mu^y + \frac{1}{2}s^\mu s^\nu\partial_\mu^y\partial_\nu^y\right)G^0(y,x')$$

$$= \frac{32}{N\pi^2}\int d^3y\, G^0(x,y)\left((b\cdot y)\left(\frac{1}{6}\partial_y^2 + \frac{3}{8y_d^2}\right) + \frac{1}{6}b^\mu\partial_\mu^y\right)G^0(y,x'). \tag{106}$$

Writing $b^\mu \partial_\mu^y = \frac{1}{2}[\partial_y^2, (b \cdot y)]$,

$$
\begin{aligned}
\delta_\epsilon G_m^a(x, x') &= \frac{32}{N\pi^2} \int d^3y \, G^0(x, y) \left( -\frac{1}{12} \{\mathcal{L}_y, b \cdot y\} + \frac{b \cdot y}{2y_d^2} \right) G^0(y, x') \\
&= -\frac{8}{3N\pi^2} b \cdot (x + x') G^0(x, x') + \frac{16}{N\pi^2} \int d^3y \, G^0(x, y) \frac{b \cdot y}{y_d^2} G^0(y, x'). \quad (107)
\end{aligned}
$$

Let's define

$$
\begin{aligned}
G_{m,nconf}^a(x, x') &= G_{m,nconf}^{a,1}(x, x') + G_{m,nconf}^{a,2}(x, x'), \\
G_{m,nconf}^{a,1}(x, x') &= -\frac{\eta}{2} \log(4x_d x_d' \Lambda^2) G_0(x, x'), \quad (108) \\
G_{m,nconf}^{a,2}(x, x') &= \frac{8}{N\pi^2} \int d^3y \, G^0(x, y) \frac{\log \Lambda' y_d}{y_d^2} G^0(y, x').
\end{aligned}
$$

Here

$$
\eta \approx \frac{8}{3N\pi^2} \quad (109)
$$

is the anomalous dimension of $\phi$ in the large-$N$ limit: $\Delta_\phi = \frac{1}{2}(d - 2 + \eta)$. $\Lambda$ and $\Lambda'$ are UV cut-offs (for future convenience, we allow them to differ by a constant factor). It is easy to check that $G_{m,nconf}^a$ has the same transformation properties (105), (107) under scale and special conformal transformations. We, therefore, conclude

$$
G_m^a(x, x') = G_{m,nconf}^a(x, x') + G_{m,conf}^a(x, x'), \quad (110)
$$

where $G_{m,conf}^a(x, x')$ transforms as a two-point function of a conformal scalar with dimension $1/2$, i.e.

$$
G_{m,conf}^a(x, x') = \frac{1}{(4zz')^{1/2}} g^1(v), \quad (111)
$$

with $g^1$ - as yet an undetermined function.

We now proceed to determine $g^1$. We have

$$
\mathcal{L}_x \mathcal{L}_{x'} G_m^a(x, x') = -\Sigma_m^a(x, x'). \quad (112)
$$

It is easy to check that $\mathcal{L}_x \mathcal{L}_{x'} G_{m,nconf}^a(x, x') = 0$ up to contact terms. Recalling Eq. (72), we then have

$$
(\mathcal{D}^2 g^1)(v) = \frac{8}{N\pi^3} \frac{v^2}{(v^2 - 1)^{5/2}}. \quad (113)
$$

This equation can be integrated to give:

$$
\begin{aligned}
g^1(v) &= \frac{2}{N\pi^3} \left( \frac{1}{3\sqrt{v^2 - 1}} \left( 1 + v \log \frac{v + 1}{v - 1} \right) + Li_2(1 - u) + Li_2(-u) + \log u \cdot \log(u + 1) + \frac{\pi^2}{12} \right) \\
&\quad + c_1 q^0(v) + c_2 g^0(v), \qquad u = \sqrt{\frac{v + 1}{v - 1}}, \quad (114)
\end{aligned}
$$

with

$$
q^0(v) = \frac{1}{8\pi} \left( 1 - \frac{v}{\sqrt{v^2 - 1}} \right) \log(v + \sqrt{v^2 - 1}), \quad (115)
$$

and $Li_2$ - the dilogorthim function. Note that Eq. (113) is a fourth order differential equation, so, in principle, there are two more independent homogeneous solutions: $c_3 \frac{v}{\sqrt{v^2-1}} \log(v + \sqrt{v^2 - 1})$ and a constant $c_4$. However, both of these don't decay as $v \to \infty$

($\rho \to \infty$), so they have wrong asymptotics for $G_m^a$ (and would violate clustering). We further note that the $c_2$ term in (114) can be incorporated into a redefinition of the cut-off $\Lambda$ in (108). Likewise, the $c_1$ term can be incorporated into the redefinition of the cut-off $\Lambda'$ in (108). Indeed, we have $\mathcal{D}q^0(v) = g^0(v)$, which means that the $c_1$ term contributes $\frac{c_1}{z^2}\delta^3(x - x')$ to $\Sigma_m^a(x, x')$. Since $G_{m,nconf}^{a,2}(x, x')$ contributes

$$\Sigma_{m,nconf}^{a,2}(x, x') = -\frac{8}{\pi^2 N}\frac{\log \Lambda' z}{z^2}\delta^3(x - x') \tag{116}$$

to $\Sigma_m^a(x, x')$ we see that the $c_2$ term can, indeed, be eliminated by a redefinition of $\Lambda'$. Thus, we set $c_1 = c_2 = 0$ from here on.

We next turn our attention to $\Sigma_m^b$ and $\Sigma_m^c$. As already remarked, both of these can be expressed in terms of $G_m^a(x, x')$, Eqs. (96). We note that the contribution to $\Sigma_m^b$ and $\Sigma_m^c$ from $G_{nconf}^{a,2}$ cancels with $\Sigma_{m,nconf}^{a,2}$. In fact, as was shown in Refs. [28], this is true for any contribution to $\Sigma_m^a$ that behaves as

$$\delta\Sigma_m^a(x, x') = U(z)\delta^3(x - x'), \tag{117}$$

with $U$ a function of $z$ and the cut-off only. Indeed, the contribution of such $\delta\Sigma_m^a$ to $\Sigma_m^b$ and $\Sigma_m^c$ is

$$\begin{aligned}\delta\Sigma_m^b(x, x') + \delta\Sigma_m^c(x, x') &= -\frac{N-1}{2}\delta^3(x - x')\int d^3y\, d^3w\, D_{\lambda\lambda}^0(x, y)\Pi(y, w)U(w_d) \\ &= -U(z)\delta^3(x - x'), \end{aligned} \tag{118}$$

where $\Pi$ is given by Eq. (84). Thus, from here on, when computing $G_m$, we drop $G_{m,nconf}^{a,2}$ and its contributions to $\Sigma_m^b$ and $\Sigma_m^c$ (we will place a hat on these quantities to denote this fact). As we will show below, the remaining contribution to $\Sigma_m^b + \Sigma_m^c$ vanishes. First, however, we note that to $O(1/N)$, $\Sigma_m^b(x, x') + \Sigma_m^c(x, x') = \langle i\delta\lambda(x)\rangle\delta^3(x - x')$, so $\langle i\delta\lambda(x)\rangle = \frac{8}{\pi^2 N}\frac{\log \Lambda' z}{z^2}$ and

$$\langle i\lambda(x)\rangle = \frac{3}{4z^2}\left(1 + \frac{32}{3\pi^2 N}\log \Lambda' z + O(N^{-2})\right). \tag{119}$$

This agrees with $\langle i\lambda(x)\rangle \sim z^{-\Delta_\lambda}$, with $\Delta_\lambda = 2 - \frac{32}{3\pi^2 N}$.

With these remarks, let's show that the remaining contributions to $\hat{\Sigma}_m^b + \hat{\Sigma}_m^c$ vanish. We begin with $\hat{\Sigma}_m^b$. We need $\hat{G}_m^a(x, x')$ at coincident points $x \to x'$. We observe,

$$g^1(v) \to \frac{2}{\pi^3 N}\left(\frac{1}{3\sqrt{2(v-1)}}\left(1 - \log\frac{v-1}{2}\right) - \frac{\pi^2}{4}\right), \quad v \to 1^+, \tag{120}$$

and

$$\hat{G}_m^a(x, x') \to \frac{1}{3\pi^3 Ns}(1 - 2\log s\Lambda) - \frac{1}{4\pi Nz}\left(1 - \frac{8}{3\pi^2}\log 2\Lambda z\right), \quad s = |x - x'| \to 0.$$

The first divergent term in $\hat{G}_m^a$ above contributes to a shift of the critical value of $g_{bulk}$ and so can be dropped, so

$$\hat{\Sigma}_m^b(x, x') = \frac{(N-1)\eta}{8\pi}\delta^3(x - x')\int d^3y\, D_{\lambda\lambda}^0(x, y)\frac{\log 2\Lambda y_d}{y_d}, \tag{121}$$

where we have used Eq. (95).

As for $\hat{\Sigma}^c_m(x, x')$, we can move $\mathcal{L}_{w'}$ in Eq. (96) onto $\sigma_0(w')$ keeping track of the boundary terms:

$$\int d^3 w' \, \mathcal{L}_{w'} \hat{G}^a_m(w, w') \sigma_0(w') = -\frac{a^0_\sigma}{\sqrt{2w'_d}} \left(\partial_{w'_d} + \frac{1}{2w'_d}\right) \hat{G}^a_m(w_d, w'_d, p = 0)\Big|^\infty_{w'_d = 0}. \tag{122}$$

We have contributions to the right-hand-side from $G^a_{m,conf}$ and $G^{a,1}_{m,nconf}$. We will see the contribution for $G^a_{m,conf}$ vanishes. On the other hand, for $G^{a,1}_{m,nconf}$ the contribution from the lower bound $w'_d = 0$ is non-zero. We have

$$G^0(z, z', p = 0) = \frac{1}{2} \frac{z^{3/2}}{z'^{1/2}}, \qquad z < z', \tag{123}$$

so

$$\int d^3 w' \, \mathcal{L}_{w'} \hat{G}^{a,1}_{m,nconf}(w, w') \sigma_0(w') = -\frac{\eta a^0_\sigma}{4\sqrt{2w_d}} (2\log(4w_d a' \Lambda^2) + 1), \tag{124}$$

where we have cut-off the integral over $w'_d$ at $w'_d = a'$. On the other hand,

$$\hat{G}^a_{conf}(z, z', p = 0) = 2\pi \sqrt{zz'} \int^\infty_{\frac{z^2 + z'^2}{2zz'}} dv \, g^1(v). \tag{125}$$

The lower bound of the integral diverges for $z' \to 0$ and $z' \to \infty$. Now,

$$g^1(v) \to \frac{8}{9N\pi^3 v^3}, \qquad v \to \infty. \tag{126}$$

This means that $G^a_{conf}(z, z', p = 0) \propto \frac{z'^{5/2}}{z^{3/2}}$ for $z' \to 0$ and $G^a_{conf}(z, z', p = 0) \propto \frac{z^{5/2}}{z'^{3/2}}$ for $z' \to \infty$. Hence, there is no contribution from either the upper or lower bound in Eq. (122). Thus, substituting (124) into Eq. (96),

$$\hat{\Sigma}^c_m(x, x') = -\hat{\Sigma}^b_m(x, x'). \tag{127}$$

Thus, we have our final result,

$$G_m(x, x') = \frac{\Lambda^{-\eta}}{(4zz')^{(1+\eta)/2}} \left(g^0(v) + g^1(v)\right), \tag{128}$$

where, again, $g^1(v)$ is evaluated with $c_1 = c_2 = 0$. We are now ready to extract $b_t$ and $a_\sigma$ to $O(1/N)$. First, let's look at the short-distance behavior $G_m(x, x')$ for $x \to x'$. From (120),

$$G_m(x, x') = \frac{\Lambda^{-\eta}}{4\pi} \left(\frac{1}{s^{1+\eta}} \left(1 + \frac{\eta}{2}\right) - 2\left(1 + \frac{1}{N}\right) \frac{1}{(2z)^{1+\eta}}\right), \qquad s = |x - x'| \to 0. \tag{129}$$

Thus, to normalize $\phi(x)$,

$$\phi_{norm}(x) = \Lambda^{\eta/2} \sqrt{4\pi} \left(1 - \frac{\eta}{4}\right) \phi(x), \tag{130}$$

and from Eq. (81),

$$a^2_{\sigma,norm} = 2(N + 1)\left(1 - \frac{\eta}{2}\right), \tag{131}$$

where we have taken the proper normalization of $\phi$ into account. As for $b_t$, as already explained, $G_\sigma(x, x')$ falls off faster than $G(x, x')$ for $\rho \to \infty$, thus, $G_m(x, x') \to \frac{N-1}{N} G(x, x')$, as $\rho \to \infty$. Further, from (126), $g^1(v) \sim v^{-3}$ as $v \to \infty$, while $g^0(v) \sim v^{-2}$ as $v \to \infty$. Thus,

$$G(x, x') \overset{\rho \to \infty}{\to} \left(1 + \frac{1}{N}\right) \frac{\Lambda^{-\eta}}{(4zz')^{(1+\eta)/2}} g^0(v), \tag{132}$$

$$b^2_{t,norm} = \frac{1}{4}\left(1 + \frac{1}{N}\right)\left(1 - \frac{\eta}{2}\right), \tag{133}$$

i.e.

$$\frac{a^2_{\sigma,norm}}{b^2_{t,norm}} = 8N + O(N^{-1}). \tag{134}$$

It is not clear from our calculation above if there is any deep reason for cancellation of the first correction to $a^2_\sigma/b^2_t$. For completeness, we present the $O(N)$ trace of the fully normalized two-point function:

$$\begin{aligned}
\langle\phi^a(x)\phi^a(x')\rangle_{norm} &= \frac{N}{(4zz')^{\Delta_\phi}}\left[\frac{a^2_{\sigma,norm}}{N} + 2\left(1 - \frac{4}{3\pi^2 N}\right)\left(\frac{v}{\sqrt{v^2-1}} - 1\right)\right. \\
&+ \frac{8}{N\pi^2}\left(\frac{1}{3\sqrt{v^2-1}}\left(1 + v\log\frac{v+1}{v-1}\right) + Li_2(1-u) + Li_2(-u)\right. \\
&+ \left.\left.\log u \cdot \log(u+1) + \frac{\pi^2}{12}\right)\right], \qquad u = \sqrt{\frac{v+1}{v-1}}. \tag{135}
\end{aligned}$$

For completeness, we also compute the correlator $G_\sigma(x,x')$ to leading order in $N$. We begin with Eq. (87) and apply $\mathcal{L}_x\mathcal{L}_{x'}$ to it:

$$\mathcal{L}_x\mathcal{L}_{x'}G^0_\sigma(x,x') = -\sigma_0(x)D^0_{\lambda\lambda}(x,x')\sigma_0(x'), \quad x \neq x'. \tag{136}$$

Writing

$$G^0_\sigma(x,x') = \frac{1}{\sqrt{4zz'}}g^0_\sigma(v), \tag{137}$$

we have

$$\mathcal{D}^2 g^0_\sigma(v) = \frac{8v}{(v^2-1)^2}. \tag{138}$$

Integrating this equation

$$g^0_\sigma(v) = \frac{2}{\pi^3}\left[-\log u - \frac{v}{\sqrt{v^2-1}}\left(Li_2(-u) + Li_2(1-u) + \log u \cdot \log(u+1) + \frac{\pi^2}{12}\right)\right]. \tag{139}$$

Here we've chosen the four integration constants so that $g^0_\sigma(v)$ decays as $v^{-3}$ for $v \to \infty$ and $g^0_\sigma(v) \to \frac{1}{2\pi\sqrt{2(v-1)}}$ for $v \to 1$. After normalization, we have

$$\begin{aligned}
\langle\phi_N(x)\phi_N(x')\rangle_{norm} &= \frac{1}{(4zz')^{\Delta_\phi}}\left[a^2_\sigma + \frac{8}{\pi^2}\left(-\log u - \frac{v}{\sqrt{v^2-1}}\left(Li_2(-u) + Li_2(1-u)\right.\right.\right. \\
&+ \left.\left.\left.\log u \cdot \log(u+1) + \frac{\pi^2}{12}\right)\right)\right], \qquad u = \sqrt{\frac{v+1}{v-1}}. \tag{140}
\end{aligned}$$

## B  $N = 2$: renormalization of velocity.

Here we analyze the problem in section 6 in the case when there is a mismatch between the surface and bulk velocities. We have

$$S = S_{ordinary} + \frac{1}{2g}\int dx\,d\tau\left(\frac{1}{v_s}(\partial_\tau\varphi)^2 + v_s(\partial_x\varphi)^2\right) - \frac{\tilde{s}v_b}{2}\int dx\,d\tau\left(e^{i\varphi}\hat{\phi}^* + e^{-i\varphi}\hat{\phi}\right). \tag{141}$$

When $\tilde{s} = 0$, we normalize

$$
\begin{aligned}
\langle e^{i\varphi(x,\tau)}e^{-i\varphi(0)}\rangle &= \frac{1}{(x^2 + v_s^2\tau^2)^{g/2\pi}}, \\
\langle \hat{\phi}(x,\tau)\hat{\phi}^*(0)\rangle &= \frac{1}{(x^2 + v_b^2\tau^2)^{\Delta_{\hat{\phi}}}}.
\end{aligned}
\tag{142}
$$

If we set $v_s = 1$ we have the OPE:

$$
e^{i\varphi(x,\tau)}e^{-i\varphi(0)} \sim \frac{1}{x^{g/2\pi}}\left(1 + ix^\mu\partial_\mu\varphi(0) - \frac{1}{4}x^2(\partial_\rho\varphi(0))^2 - \frac{g}{2}x^\mu x^\nu T_{\mu\nu}(0) + \dots\right),
\tag{143}
$$

with the energy-momentum tensor,

$$
T_{\mu\nu} = \frac{1}{g}\left(\partial_\mu\varphi\partial_\nu\varphi - \frac{1}{2}\delta_{\mu\nu}(\partial_\rho\phi)^2\right).
\tag{144}
$$

Here we have omitted derivatives of $\partial_\mu\varphi$ on the right-hand-side of (143). Restoring $v_s$,

$$
\begin{aligned}
e^{i\varphi(x,\tau)}e^{-i\varphi(0)} \sim \frac{1}{(x^2 + v_s^2\tau^2)^{g/2\pi}}\Big(&1 + i(x\partial_x\varphi(0) + \tau\partial_\tau\varphi(0)) \\
&- \frac{1}{2}(x^2(\partial_x\varphi)^2 + \tau^2(\partial_\tau\varphi)^2 + 2x\tau\partial_x\varphi\partial_\tau\varphi)\Big).
\end{aligned}
\tag{145}
$$

Thus, in an RG step we generate:

$$
\begin{aligned}
\delta S &= \frac{\tilde{s}^2 v_b^2}{8}\int dx\,d\tau \int_{a^2 < x'^2 + v_b^2\tau'^2 < a^2(1+2d\ell)} dx'\,d\tau'(x'^2(\partial_x\varphi(x))^2 + \tau'^2(\partial_\tau\varphi(x))^2)\frac{1}{(x'^2 + v_s^2\tau'^2)^{\frac{g}{4\pi}}(x'^2 + v_b^2\tau'^2)^{\Delta_{\hat{\phi}}}} \\
&= d\ell\,\tilde{s}^2\int dx\,d\tau\left(\frac{v_s}{2}A(v_s/v_b)(\partial_x\varphi)^2 + \frac{1}{2v_s}B(v_s/v_b)(\partial_\tau\varphi)^2\right),
\end{aligned}
\tag{146}
$$

with

$$
\begin{aligned}
A(v_s/v_b) &= \frac{v_b}{4v_s}\int_0^{2\pi} d\theta\,\frac{\cos^2\theta}{(\cos^2\theta + \frac{v_s^2}{v_b^2}\sin^2\theta)^{\frac{g}{4\pi}}}, \\
B(v_s/v_b) &= \frac{v_s}{4v_b}\int_0^{2\pi} d\theta\,\frac{\sin^2\theta}{(\cos^2\theta + \frac{v_s^2}{v_b^2}\sin^2\theta)^{\frac{g}{4\pi}}}.
\end{aligned}
\tag{147}
$$

(Here, we have set $(2 - \Delta_{\hat{\phi}}) = \frac{g}{4\pi}$.) Thus,

$$
\begin{aligned}
\frac{d\tilde{s}}{d\ell} &= (2 - \Delta_{\hat{\phi}} - \frac{g}{4\pi})\tilde{s}, \\
\frac{dg}{d\ell} &= -\frac{1}{2}(A+B)\tilde{s}^2 g^2,
\end{aligned}
\tag{148}
$$

$$
\frac{d(v_s/v_b)}{d\ell} = \frac{1}{2}(A-B)g\tilde{s}^2\frac{v_s}{v_b}.
\tag{149}
$$

We define:

$$
u = \frac{g}{4\pi} - (2 - \Delta_{\hat{\phi}}), \quad v = \sqrt{2\pi(A+B)}(2 - \Delta_{\hat{\phi}})\tilde{s}.
\tag{150}
$$

Then

$$
\begin{aligned}
\frac{dv}{d\ell} &= -uv + O(v^3), \\
\frac{du}{d\ell} &= -v^2, \\
\frac{d(v_s/v_b)}{d\ell} &= \frac{1}{2-\Delta_{\hat{\phi}}}\frac{A-B}{A+B}v^2\frac{v_s}{v_b}.
\end{aligned}
\tag{151}
$$

We see that to the present order, the flow of $v_s$ does not affect the flow of $u$ and $v$. Thus, we have the same separatrix:

$$
u(\ell) = v(\ell) = \frac{v}{1+v\ell}.
\tag{152}
$$

Then, integrating the RG equation for $v_s/v_b$,

$$
\frac{v_s(\ell)}{v_b} \approx \left(1 + \frac{A-B}{(A+B)(2-\Delta_{\hat{\phi}})}, \frac{v^2\ell}{1+v\ell}\right)\frac{v_s}{v_b}.
\tag{153}
$$

Thus, we see that at the transition, the surface velocity $v_s$ renormalizes slightly, but does not sync with the bulk velocity.

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
