# Peer review of "Boundary criticality of the O(N) model in d = 3 critically revisited"

_SciPost Physics, doi:SciPost Phys. 12, 131 (2022)_

## Round 1 · Referee Report · Anonymous · 2021-12-9

Strengths

1- Clear review of the status of the problem
2- Important advance in the field
3- Broad discussion of classical and quantum models

Weaknesses

none

Report

The author studies the classical boundary critical behavior of the O(N) universality class in three dimensions.
This is a mature subject in statistical physics, yet it has received a lot of attention recently in the context of quantum spin models.
Using field-theoretical methods, the paper shows that for N <= Nc the O(N) model hosts a new boundary universality class, where the two-point function of the order parameter decays as a power of the logarithm of the distance.
The threshold Nc cannot be determined by field-theory calculations, nevertheless one can show that Nc >= 2.
It should be mentioned that Monte Carlo simulations appeared after the first preprint version of this manuscript have confirmed this picture.
The author also discusses various scenarios for the boundary critical behavior of quantum models, related to the O(N) universality class.

This is a nicely written and very interesting paper which has an important impact on the field.
I fully recommend its publication.

---

## Round 1 · Referee Report · Anonymous · 2021-12-21

Report

The paper discusses the boundary critical behavior of the O(N) model. This is an old subject, and yet the author points out that a region in the phase diagram has not been understood. Amusingly, the missing piece of the puzzle is even referred to in Cardy's book: at page 137, when discussing the extraordinary transition (i.e. the phase where the boundary spontaneously break the symmetry above the bulk critical temperature) Cardy comments: "This assumes, of course, that d-1 is larger than the lower critical dimension". This crucially leaves out the case d=3, which is what the present paper focuses on.

While the problem is obvious, the solution is interesting. The topology of the phase diagram turns out to depend non-trivially on N, and showcases a region with quasi-long-range order on the boundary, below a certain critical value Nc. While Nc is not determined in this work, it is certainly larger than 2. The correlator of the order parameter decays like a power of a logarithm: these multiplicative logarithms are typical when the large distance limit is controlled by a marginally irrelevant coupling, like, as the authors shows, in this case.

The paper is important in many respects: first and foremost, it fills a gap in our understanding of a basic statistical model with many applications. Furthermore, since this gap happened to be about three dimensional physics, the results presented here might be experimentally relevant. Finally, the renormalization group analysis leaves open various possibilities for the phase diagram above Nc: it is likely that this will foster further research on the subject.

The paper is well written and clearly organized. I am happy to recommend it for publication as is.

---

## Editorial Decision

published